# Emergence of periodic circumferential actin cables from the anisotropic fusion of actin nanoclusters during tubulogenesis

Sayaka Sekine [1,2,7] ✉, Mitsusuke Tarama [3,4,7] ✉, Housei Wada[1], Mustafa M. Sami[1,5], Tatsuo Shibata [3] & Shigeo Hayashi[1,6]

The periodic circumferential cytoskeleton supports various tubular tissues. Radial expansion of the tube lumen causes anisotropic tensile stress, which can be exploited as a geometric cue. However, the molecular machinery linking anisotropy to robust circumferential patterning is poorly understood. Here, we aim to reveal the emergent process of circumferential actin cable formation in a *Drosophila* tracheal tube. During luminal expansion, sporadic actin nanoclusters emerge and exhibit circumferentially biased motion and fusion. RNAi screening reveals the formin family protein, DAAM, as an essential component responding to tissue anisotropy, and non-muscle myosin II as a component required for nanocluster fusion. An agent-based model simulation suggests that crosslinkers play a crucial role in nanocluster formation and cluster-to-cable transition occurs in response to mechanical anisotropy. Altogether, we propose that an actin nanocluster is an organizational unit that responds to stress in the cortical membrane and builds a higher-order cable structure.

Tubular networks are essential for supporting organism life by permitting the circulation of vital nutrients and air. A key feature of functional epithelial tubules is their ability to resist the expansion force of circulating liquids or gases. One mechanical solution for robustly maintaining the tube diameter while allowing flexibility in tube curvature is to introduce periodic circumferential actin cables. Such structures are generally found in multiple systems at different length scales ranging from the nanometer to the micrometer scale, including neuronal processes[1], vertebrate blood vessels[2], the hypoderm of *Caenorhabditis elegans* embryos[3], and *Drosophila* trachea[4]. Plant vessels are surrounded by circumferential cortical microtubules[5].

*Drosophila* trachea is a respiratory system that enables gas exchange through a continuous network of epithelial tubules[6,7]. At

embryonic stage 15, tracheal cells begin to secrete the extracellular matrix and liquid into the lumen, causing remarkable expansion of the luminal diameter[8–10]. Concomitantly, regularly arrayed supracellular circumferential actin cables transiently appear beneath the apical membrane (Fig. 1a, b)[4]. Many mutations that perturb the actin cable formation also cause irregularities in the taenidial fold covering the luminal surface of the mature trachea, suggesting that the actin cables might be essential for patterned cuticle secretion into the lumen[11,12]. The formin actin nucleator protein, DAAM, functions downstream of the small GTPase, RhoA, and helps align actin cables along the circumferential direction[4]. As radial expansion of the tubule causes anisotropic tensile stress on the membrane[13], our group previously proposed that local membrane anisotropy acts as a geometric cue for the circumferential alignment of actin cables[14]. Based on the

[1]Laboratory for Morphogenetic Signaling, RIKEN Center for Biosystems Dynamics Research, Kobe, Japan. [2]Laboratory for Histogenetic Dynamics, Graduate School of Life Sciences, Tohoku University, Sendai, Japan. [3]Laboratory for Physical Biology, RIKEN Center for Biosystems Dynamics Research, Kobe, Japan. [4]Department of Physics, Faculty of Science, Kyushu University, Fukuoka, Japan. [5]Physics and Biology Unit, Okinawa Institute of Science and Technology Graduate University, Okinawa, Japan. [6]Kobe University Graduate School of Science, Kobe, Japan. [7]These authors contributed equally: Sayaka Sekine, Mitsusuke Tarama. ✉e-mail: sayaka.sekine.d7@tohoku.ac.jp; tarama.mitsusuke@phys.kyushu-u.ac.jp

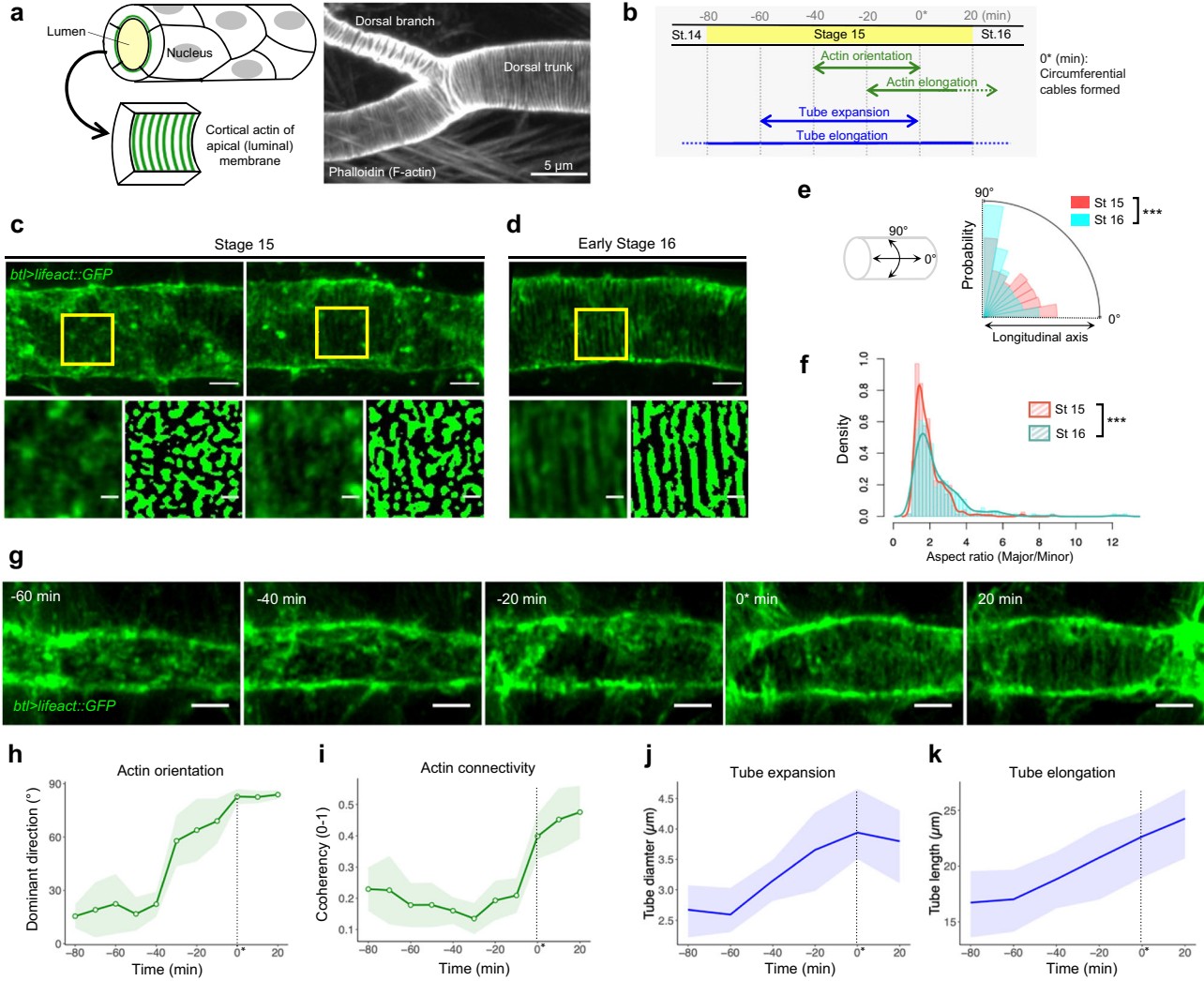

**Fig. 1 | Emergence of actin nanoclusters in the apical cortex during tracheal tube expansion. a** Schematic of the tracheal tube constituted by multiple cells. The intercellular periodic circumferential actin cables appear beneath the apical (luminal) membrane. The right image shows a representative image of F-actin pattern at stage 16 visualized by phalloidin staining. Three experiments were repeated independently. Scale bar, 5 μm. **b** Time course of cortical actin orientation, actin cable elongation, tracheal tube expansion, and tube elongation at stage 15. The time point when the circumferential cables formed was defined as 0* min. **c, d** Actin pattern visualized by lifeact::GFP in the fixed embryos at stages 15 and 16, respectively. Two representative images for stage 15 are shown in (**c**). Lower square images indicate a magnified view of the yellow square (left) and its binarized image (right). Scale bars: 2 μm (upper) and 0.5 μm (lower). Eight and seven independent tracheal tubes were observed for stage 15 and stage 16, respectively. **e, f** Comparison of the cluster-fitted ellipses between stage 15 (*n* = 356, *N* = 22) and stage 16 (*n* = 271, *N* = 21) with aspect ratio (**f**), and angle of the major axis of clusters (**e**, 0° corresponds to longitudinal axis whereas 90° indicates circumferential axis of

the tube). For statistical analyses, two-sided two-sample Kolmogorov–Smirnov test was performed for **f** (*P* = 3.17e-4), and two-sided Watson's two-sample test of homogeneity was performed for **e** (*P* < 0.001). ***P* < 0.001. See "Cluster analysis" for details. **g** Time-lapse images showing the tracheal tube of living embryos expressing lifeact::GFP in the tracheal cells. Scale bar, 2 μm. **h, i** Transition of dominant direction and coherency of the actin pattern every 10 min. Coherency indicates how the gradient tensor of the actin intensity distribution pattern is aligned toward the dominant direction. The coherency is used as a measure of the connectivity of the nanoclusters in this study ("OrientationJ analysis"). The mean (green line) and SD (light green region) are indicated. *n* = 6 (−80, −70, −60, −50, −40 min), 7 (−30 min), 9 (−20, −10, 0 min), 8 (10 min), or 5 (20 min) biologically independent samples. **j, k** Transition of tube diameter and length of the dorsal trunk. The mean (blue line) and range of values (light blue region) are indicated. *n* = 6 biologically independent samples. The dotted vertical lines in **h–k** indicate when the circumferential cables are formed (0* min). Source data are provided as a Source Data file.

phenomenological active gel model, the local anisotropic friction between the actomyosin gel and the membrane permits the self-organization of periodic actin cables[14]. However, the molecular machinery linking anisotropy to the robust organization of the underlying cortical actin network remains to be elucidated.

Here, we investigate the process of periodic circumferential pattern formation in the tracheal tube of the embryonic stage. The live imaging with high spatiotemporal resolution reveals the dynamic fusion and fission of sporadic actin nanoclusters in the luminal cortex. Circumferentially biased stabilization of the fusion leads to periodic

cable formation. Candidate RNAi screen isolated a formin family protein DAAM, actin cross-linker α-Actinin and Zasp52, and non-muscle Myosin II as essential molecules organizing the nanocluster. An agent-based model reproduces the whole process of the nanocluster formation and the subsequent cluster-to-cable transition, suggesting that this process can be explained by self-organization. Altogether, we conclude that the actin nanoclusters are organizational units that respond to local membrane anisotropy and develop into periodic circumferential cables. This anisotropy sensing model can be applied to a broader context of tissues or cells.

## Results

### Actin nanoclusters appeared in the luminal cortex of tracheal cells

To study the molecular mechanism of actin cable formation, we first observed the transition of cortical actin patterns at several embryonic stages. As the process is highly dynamic in deep tissue and the size of the nascent actin cables was close to the diffraction limit, we used Airyscan which enables super-resolution imaging based on deconvolution and the pixel reassignment principle[15–17]. Lifeact::GFP was specifically expressed in tracheal cells, and the luminal cortex of fixed embryos was imaged. At embryonic stage 14, junctional and sparsely clustered actin was observed in the apical cortex (Supplementary Fig. 1a). At stage 15 of tube expansion, small actin clusters emerged and covered the cortex (Fig. 1c, left). The mode value of the cluster size estimated by elliptic fitting was about 140 nm on the minor axis and 210 nm on the major axis (Supplementary Fig. 1b, c). The orientation of the major axis of the clusters was random (Fig. 1e). As the development proceeds in stage 15, the clusters started to exhibit circumferential elongation (Fig. 1c, right), then by early stage 16, the circumferential orientation was completed with a statistically significant increase of the aspect ratio (Fig. 1d–f, Supplementary Fig. 1b, c, and Supplementary Table 1). Actin clusters coalesced into paralleled cables with regular intervals (450 ± 13.2 nm, Supplementary Fig. 1d). The existence of the actin cluster pattern was confirmed by using other F-actin probes (Supplementary Fig. 1e). In the larval stage, the accumulation of Rho-GEF2 at the longitudinal junction was suggested to be an upstream signal of actin cable orientation[18]. Such biased accumulation of junctional RhoGEF2 was not observed at embryonic stages 15 and 16 (Supplementary Fig. 1f). Instead, we observed the circumferential orientation of the elliptical actin clusters as the first sign of anisotropic organization of the actin cytoskeleton (Fig. 1e). The size of the clusters was smaller than the similar structures reported previously, such as actin foci[19], actin patches[20,21], actin nodes[22], or actin condensates[23]. Thus, we named the structure that we found in our experiment as *actin nanoclusters*.

### Nanoclusters oriented circumferentially before cable formation

Live imaging revealed a detailed time course of the elongation and orientation of the actin nanoclusters during the tracheal tube expansion stage (Fig. 1j). OrientationJ[24] was used to estimate the connectivity and orientation of the nanoclusters. The orientation increased over 40 min with high variation (−40 min to 0 min in Fig. 1h; 0* min corresponds to the completion of circumferential alignment of the actin pattern, "OrientationJ analysis"), whereas the connectivity stayed low by −10 min (Fig. 1i). When the connectivity increased at 0* min, the variation of orientation substantially decreased (Fig. 1h, i). This measurement was supported by the skeleton analysis, which revealed that elongation of the actin nanoclusters became apparent at −20 min (Supplementary Fig. 1g). The time window of the orientation increase correlated well with the period of tube expansion, but not with tube elongation (Fig. 1h–k and Supplementary Fig. 1h). Taken together, the actin nanoclusters were oriented circumferentially when they were still short. Thereafter, the actin nanoclusters were elongated, which led to cable formation on the apical membrane (Fig. 1b).

### Biased stabilization in repetitive nanocluster fusion and fission

We hypothesized that the elongation of the nanoclusters was due to their fusion. The major lengths at Stage 16 exhibited a multimodal distribution (Supplementary Fig. 1b). To investigate this hypothesis, fast live imaging with a 0.32-s interval was performed (Fig. 2a and Supplementary Movies 1–3). The nanoclusters exhibit highly dynamic changes in shape and position, which may have involved rapid assembly and disassembly. Although the nanoclusters underwent fusion (marked with magenta in Fig. 2a) and formed elongated

clusters, they were broken down into nanoclusters in the next few frames owing to subsequent fission. Most fusions lasted no more than two frames (0.64 s), indicating that the cluster fusion was unstable (Fig. 2b). We proceeded to quantify the duration of fusion in each direction (Fig. 2b, Supplementary Fig. 2, and "Fusion stability"). Circumferential fusion tended to have a longer duration than diagonal or longitudinal fusion and became significant at 0* min (Fig. 2b). To summarize, high spatiotemporal imaging revealed that nanocluster fusion in the circumferential direction was preferentially stabilized; because of the numerous repetitions of fusion and fission, the slightly biased stabilization becomes apparent and reaches the steady-state pattern, the periodic cable pattern, while maintaining the dynamic local shape changes (Fig. 2c).

### RNAi screen for actin nanocluster-organizing molecules

We proceeded to search for the molecules required for biased actin nanocluster orientation and fusion using an RNAi screen. A total of 142 RNAi lines[25] targeting 119 genes encoding actin-binding proteins were tested for expression in tracheal cells (Supplementary Data 1). The actin pattern at early stage 16 was analyzed in 22 lines that exhibited larval lethality (Fig. 3a, b). The downregulation of 10 genes caused changes in the orientation and major length of the actin nanoclusters (Fig. 3a, b, Supplementary Fig. 3a–c, and Supplementary Table 2). Some of the genes, *DAAM*, *dia*, *form3*, and *cora*, have been previously reported to function in tracheal development[4,26–29], indicating the validity of the screen.

An interaction map of the screening-positive genes was constructed based on the physical and genetic interaction database (Molecular Interaction Search Tool)[30]. Eight of the ten screen-positive proteins were linked with additional interactors, including RhoA (Fig. 3c and Supplementary Fig. 4), which was indeed required for cable formation (Supplementary Fig. 5a, b). Notably, the "myofibril assembly complex (FC1047)" in the interaction map included the major actin cross-linker, α-actinin (Supplementary Data 2). Indeed, the *actn*[14] mutants exhibited a strong defect in actin cable formation (Fig. 3a, b).

For further analyses, we focused on three actin-binding proteins that exhibited strong orientation and elongation defects: DAAM, α-Actinin, and Zasp52. DAAM is a formin family protein that nucleates straight actin filaments in response to tensile stress[31–34]. α-Actinin is an actin cross-linker, and Zasp52 increases the stability of the binding between α-Actinin and actin filament[35,36]. The endogenously-tagged DAAM, α-Actinin, and zasp52 showed significant colocalization with actin nanoclusters at stage 15 (Fig. 3d and Supplementary Fig. 5c). In addition, the heavy chain of non-muscle myosin II (Zipper) and its upstream regulator, Rho-associated coiled-coil kinase (ROCK, Rok in *Drosophila*), colocalized with the actin nanoclusters, suggesting that they directly drive the motility of the nanoclusters (Fig. 3d and Supplementary Fig. 5c). In contrast, non-tagged GFP or RhoGEF2 did not co-localize with the actin nanoclusters, indicating that only the selected molecules were localized to the actin nanoclusters (Fig. 3d and Supplementary Fig. 5c). Most of the nanocluster-associated molecules colocalized with the actin cables at stage 16 (Supplementary Fig. 5c, d).

### Molecules for nanocluster fusion and its biased stabilization

To identify the molecules crucial for local anisotropy response, nanocluster fusion analysis was performed on fast live imaging at early stage 16 ("Fusion stability"). The duration of circumferential fusion was significantly longer than that of diagonal or longitudinal fusion in control, *zasp52* RNAi, and dominant-negative form of *myosin II* (*myosin II*[DN], *zip*[ROD] in *Drosophila*[37])-expressing tracheal cells (Fig. 3e). In contrast, the *DAAM* RNAi-expressing cells exhibited a comparable duration of fusion in all directions (Fig. 3e), which led to the formation of a labyrinth pattern of randomly connected actin

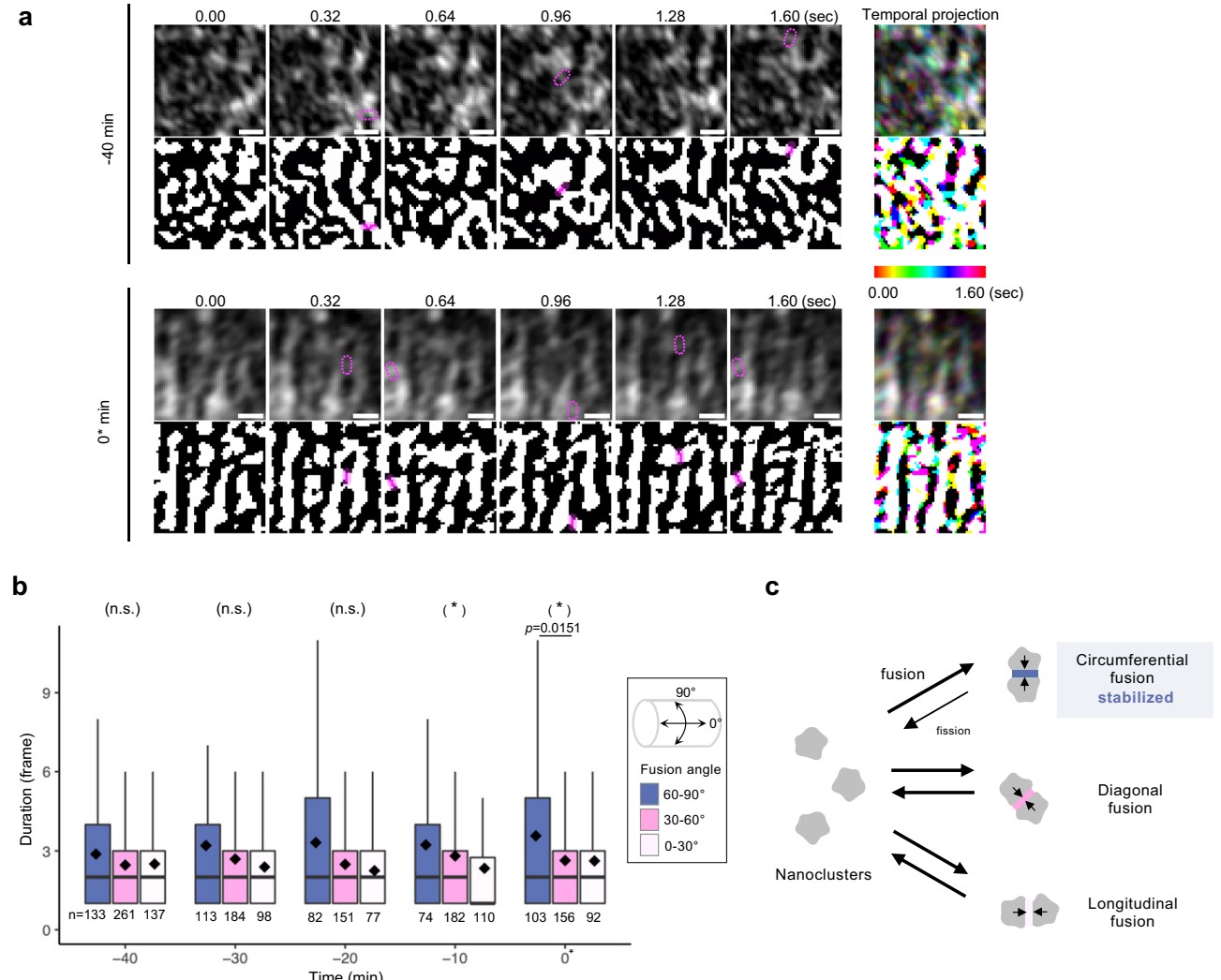

**Fig. 2 | Nanocluster fusion is more stable in the circumferential direction.**
**a** Time-lapse fluorescent images and binarized images of lifeact::GFP expressing tracheal cells with 0.32-s intervals at stage 15. The fusion sites of clusters were labeled with magenta (see Supplementary Fig. 2 and "Fusion stability"). The right-most images are the temporal projection of the six frames, color-coded with Spectrum. Upper: Actin pattern at −40 min. Actin nanoclusters change in shape and size dynamically. Lower: Actin pattern at 0* min. Although the circumferential cables become apparent, the cables are still unstable. Scale bar, 0.5 μm. **b** Duration of the fusion of the nanoclusters with respect to the orientation. The duration of circumferential fusion became significantly longer than that of the diagonal fusion at 0* min. 0° corresponds to the longitudinal axis, whereas 90° indicates the circumferential axis of the tube. The numbers of fusions (*n*) are written below. Boxplots represent the median (bar), mean (black diamond) plus minima and maxima with lower and upper quantiles. For statistical analysis, the two-sided Kruskal−Wallis test (shown in parentheses, *P* = 0.565 in −40 min, *P* = 0.322 in −30 min, *P* = 0.193 in −20 min, *P* = 0.0343 in −10 min, and *P* = 0.0382 in 0* min) followed by a pairwise comparison using the two-sided Wilcoxon rank-sum test with Bonferroni adjustment of the *P* value (shown with bar) was performed. *P* < 0.05. See "Fusion stability" for detail. **c** Model for actin nanocluster-to-cable transition. Through repetitive fusion and fission of the nanoclusters, circumferential fusion is slightly stabilized, thereby leading to circumferential cable formation in a steady state. Source data are provided as a Source Data file.

cables at late stage 16 (Fig. 3f), aligning with previous reports[4,14]. Thus, DAAM is required for the biased stabilization of circumferential fusion.

In DAAM-downregulated cells, the actin nanoclusters failed to follow the circumferential axis, despite exhibiting fusion. To further identify the molecules essential for fusion, we examined the actin cross-linker mutants at late stage 16. The loss of α-actinin and down-regulation of zasp52 were found to result in strong fusion defects (Fig. 3f). In addition, the attenuation of Myosin II by the expression of *myosin II*[DN] also caused failure in cluster fusion (Fig. 3f), consistent with the ROCK inhibitor treatment[14]. The downregulation of RhoA by the expression of RNAi or a dominant-negative form resulted in fusion failure (Supplementary Fig. 5a, b). Taken together, we conclude that crosslinkers and RhoA/ROCK-induced myosin contractility are required for the fusion of actin nanoclusters.

## The agent-based model reproduced cluster-to-cable transition
To elucidate the self-organization mechanism of the actin nanoclusters and cables, we used a bottom-up approach and developed an agent-based model based on the components identified in the experiments (Supplementary Notes)[38–40]. First, based on the measured size of the nanocluster and reported crosslinking distance of α-Actinin[41], the length and number of actin filaments in the actin nanoclusters were estimated (Fig. 4a and Supplementary Fig. 6a). For the set of estimated parameters, we determined the conditions in which actin filaments and crosslinkers self-organized into regular clusters (Fig. 4b, *1-M, 1-H*, Supplementary Fig. 7a, and Supplementary Table 3). Further analyses revealed that the clusters in the numerical simulation were formed through the competition between the effective diffusion of the fila-ments and their effective attraction due to the crosslinkers connecting the filaments (Supplementary Notes and Supplementary Fig. 7a). In

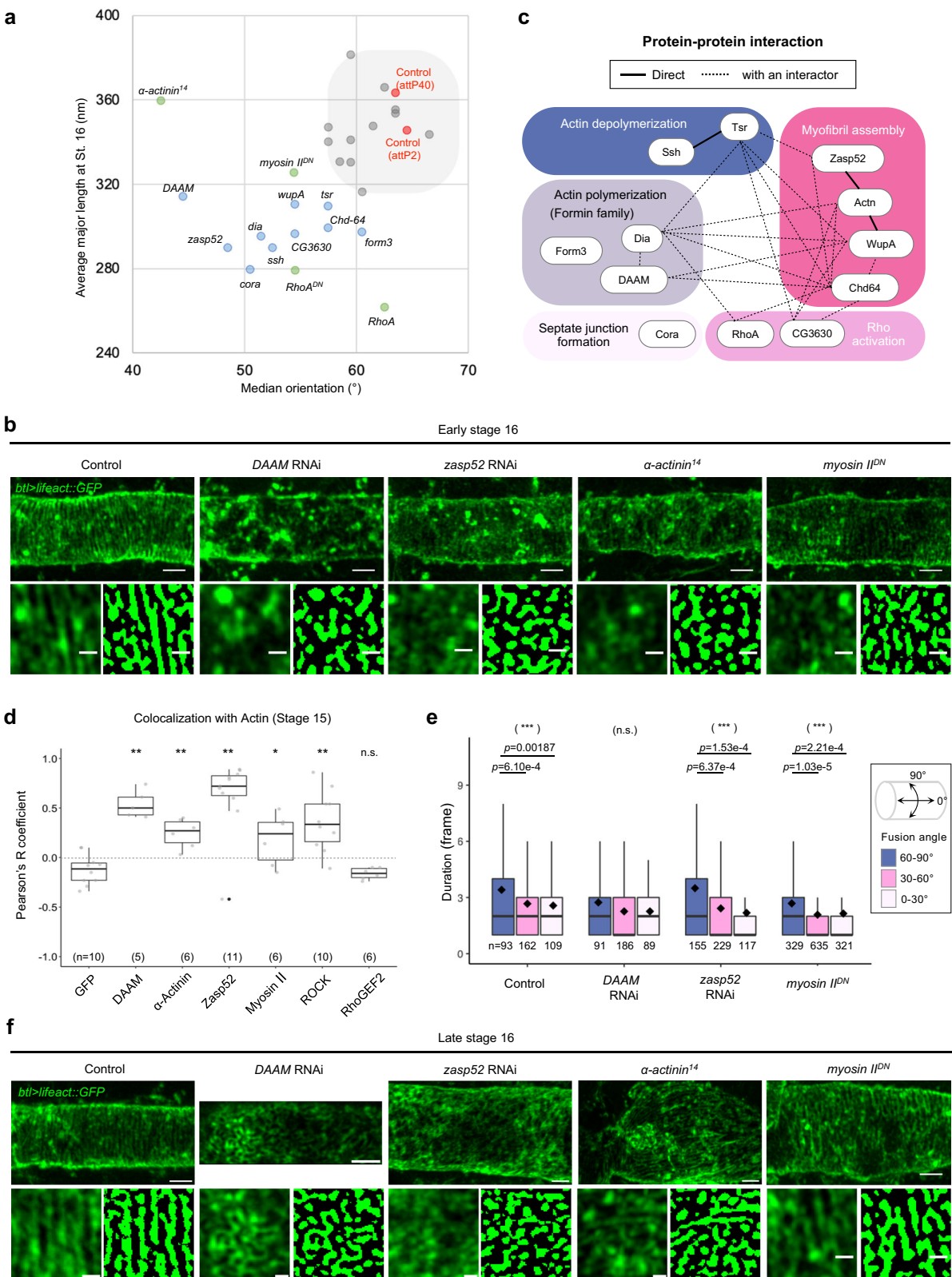

fact, the distance between the clusters changed depending on the filament turnover rate, which enhances the effective diffusion (Supplementary Fig. 7c and Supplementary Table 4). Most of the clusters were elliptical, with a minor length of ~200 nm and a major length ranging from 250 to 350 nm (Supplementary Fig. 6b, c, Supplementary Table 3, and "Cluster analysis"). These clusters were comparable to the in vivo actin nanoclusters under various genetic conditions

(Supplementary Fig. 3c and Supplementary Table 2). Thus, the simulation successfully reproduced the actin nanoclusters in a dynamic steady state, in which the individual clusters continuously exchange their constituting molecules.

Second, myosin motors were added to determine whether they led to cluster fusion. Myosin did promote the fusion of clusters and as the number of motors increases, the clusters are more connected to

**Fig. 3 | Revealing nanocluster-associated molecules that regulate cluster fusion and anisotropy response. a** Median orientation and average major length of the actin patterns are plotted for the genetically manipulated tracheal cells ("OrientationJ analysis" and "Cluster analysis"). Each circle indicates different genetic conditions: control (red), RNAi lines with mild defects (gray), RNAi lines with strong defects (blue, denoted with the target genes), and mutants found after the RNAi screen (green). See Supplementary Table 2 for details. **b** Confocal projection images and binarized images of control, RNAi-expressing, or mutant embryos at early stage 16. More than six independent tracheal tubes were observed for each genotype. Scale bars: 2 μm (upper) and 0.5 μm (lower). **c** A network of protein-protein interaction between genes, whose downregulation displayed strong phenotypes. Solid lines indicate direct interaction and dotted lines indicate indirect interaction with a single interactor. The genes are grouped by representative molecular functions. **d** Colocalization between the proteins of interest with actin (lifeact::mScarletx2 or lifeact::GFP) at stage 15. Pearson's correlation coefficient was calculated based on 2-channel images shown in Supplementary Fig. 5c. Boxplots represent median plus minima and maxima with lower and upper quantiles. *n* indicates a biologically independent sample. The two-sided Student's *t* test. $P = 1.46e\text{-}5$ (DAAM), $P = 5.43e\text{-}4$ (α-Actinin), $P = 2.71e\text{-}5$ (Zasp52), $P = 0.0321$ (Myosin II), $P = 4.67e\text{-}4$ (ROCK) or $P = 0.4951$ (RhoGEF2). *$P < 0.05$, **$P < 0.01$. **e** Duration of the nanocluster fusion with respect to the orientation in control, *zasp52* RNAi, *DAAM* RNAi, and *myosin II$^{DN}$* expressing tracheal cells at early stage 16. The numbers of fusions (*n*) are written below. Boxplots represent median (bar), mean (black diamond) plus minima and maxima with lower and upper quantiles. The two-sided Kruskal–Wallis test (parentheses, $P = 9.23e\text{-}4$ in Control, $P = 0.114$ in *DAAM* RNAi, $P = 1.41e\text{-}4$ in *zasp52* RNAi, or $P = 1.83e\text{-}5$ in *myosin II$^{DN}$*) followed by a pairwise comparison using the two-sided Wilcoxon rank-sum test with Bonferroni adjustment of the *P* value (shown with bar) was performed. ***$P < 0.001$. **f** Confocal projection images and binarized images of control, RNAi-expressing, or mutant embryos at late stage 16. Three independent tracheal tubes are observed for each genotype. Scale bars: 2 μm (upper) and 0.5 μm (lower). Source data are provided as a Source Data file.

each other (Fig. 4b, c, *3-M, 3-H*, Supplementary Fig. 7b2, and Supplementary Table 3). The clusters then underwent fusion in a random orientation, resulting in a labyrinth pattern with intervals similar to that of the *DAAM* RNAi phenotype (Fig. 3f).

Finally, anisotropic friction was introduced to the actin filaments to provide an external cue that mimicked anisotropic tensile stress[14]. Consequently, the fused actin clusters aligned toward the higher-friction direction, forming a similar pattern with the periodic actin cables (Fig. 4b, c, *4-M* and Supplementary Table 3). The interval between cables was ~0.5 μm (Supplementary Fig. 7c and Supplementary Table 4), which was comparable with the result of the in vivo analysis (Supplementary Fig. 1d). The alignment of the clusters toward the higher-friction direction is due to the difference in the relaxation time scale of the filaments, i.e., the characteristic time to react to the effective attraction and diffusion, caused by the friction asymmetry (see Supplementary Notes). This difference is expected to appear in the behavior of clusters. Indeed, the clusters showed slightly biased motion in the direction of higher friction (Fig. 4d, e and "Motion anisotropy"). This small anisotropy in the cluster motion was enhanced by the motors, which tend to bridge the clusters toward a higher-friction direction (Fig. 4d, e and Supplementary Table 5). Without a sufficient number of crosslinkers, the filaments failed to organize the clusters and exhibited no clear pattern with a specific directionality despite the anisotropic friction (compare Fig. 4b *1-L* with *2-L* and *4-L*, median orientation in Supplementary Table 3). Thus, the clustering of actin filaments plays an important role in reacting to friction anisotropy.

We further examined the process of cable formation starting from a uniform distribution of filaments in the simulation result most similar to the in vivo pattern (with motors and anisotropic friction, Fig. 4b *4-M*, and Supplementary Movie 4). The orientation first increased, followed by an increase in connectivity (Fig. 4f), which is consistent with the experimental results (Fig. 1h, i). In conclusion, the simulation demonstrated that the elliptical actin nanoclusters react to friction anisotropy and elongate in the direction of higher friction, developing into regular cables.

### Motion anisotropy of nanoclusters in vivo

The simulation results revealed that the actin nanocluster movement was biased in the higher-friction direction during cable formation (Fig. 4d, e). To determine whether this finding holds true in vivo, we tracked the motion of actin nanoclusters captured by fast imaging. The analysis revealed that the relatively small and round actin nanoclusters tended to move more in the circumferential direction from −30 min (Fig. 5a and Supplementary Table 5). This tendency was also captured by particle image velocimetry (PIV) analysis. The isotropic actin motion at the beginning becomes dominated by circumferential motion after −10 min (Fig. 5b, c).

As the stability of the circumferential fusion of the nanoclusters was strongly dependent on the DAAM (Fig. 3e), we determined whether DAAM regulated the motion of the nanoclusters. As expected, motion anisotropy analysis and PIV revealed that the motion of actin nanoclusters in the *DAAM* RNAi trachea was isotropic (Fig. 5d–f and Supplementary Table 5). Next, we investigated nanocluster motion under the downregulation of crosslinkers. The nanoclusters in *zasp52* RNAi cells lost their anisotropic motion (Fig. 5d–f and Supplementary Table 5); this is similar to the simulation results, which did not display an anisotropic pattern with a low number of crosslinkers (Fig. 4b, *2-L* and *4-L*). The excessively high turnover rate of crosslinkers might cause the diffusion of actin filament from nanoclusters thus failing to exhibit the anisotropic motion in the verified time scale (Fig. 5g). Finally, the requirement of Myosin II in the motion anisotropy was checked. In contrast to the knockdown of the other two components, the anisotropy was retained but at a lower degree in Myosin II downregulation (Fig. 5d–f). Hence, in addition to DAAM, the cross-linker-dependent clustering of actin filaments contributes to the anisotropy-responding machinery, whereas myosin II plays a role in cluster fusion.

## Discussion

In this study, we demonstrated that actin nanoclusters act as organizational units for the dynamic behavior of cortical actin and the formation of periodic circumferential actin cables during *Drosophila* tracheal tube development by fast live imaging, genetic manipulation, and theoretical simulations. By putting the following three factors together: cross-linker-dependent nanocluster self-organization, myosin II-dependent nanocluster fusion, and anisotropy response, the periodic circumferential cables are formed in silico (Fig. 5g). The simulation experiments omitting one or more of the three factors phenocopied the experimental results in vivo. First, the simulation with the low number of crosslinkers (Fig. 4b, *4-L*) exhibited an isotropic pattern (Supplementary Table 3), which was similar to the downregulation of the cross-linker, *zasp52*, that showed orientation defect (Fig. 3a). Although the nanocluster fusion is stabilized anisotropically in *zasp52* RNAi (Fig. 3e), the weaker crosslinking might cause too frequent turnover of actin filaments and thus fail to maintain the anisotropic higher-order cable structure. Second, the absence of motors resulted in isolated clusters in the model. As the number of motors increases, the clusters are more connected to each other (Supplementary Fig. 7b2), consistent with the experimental results using *myosin II$^{DN}$* (Fig. 3b, f). Third, when the system is isotropic, the direction of fusion is random resulting in a labyrinth pattern, which was comparable with the *DAAM* RNAi phenotype (Fig. 3f). Previous study reported that deformation of the cell membrane induces a rapid increase in cytoplasmic G-actin resulting in the positive regulation of

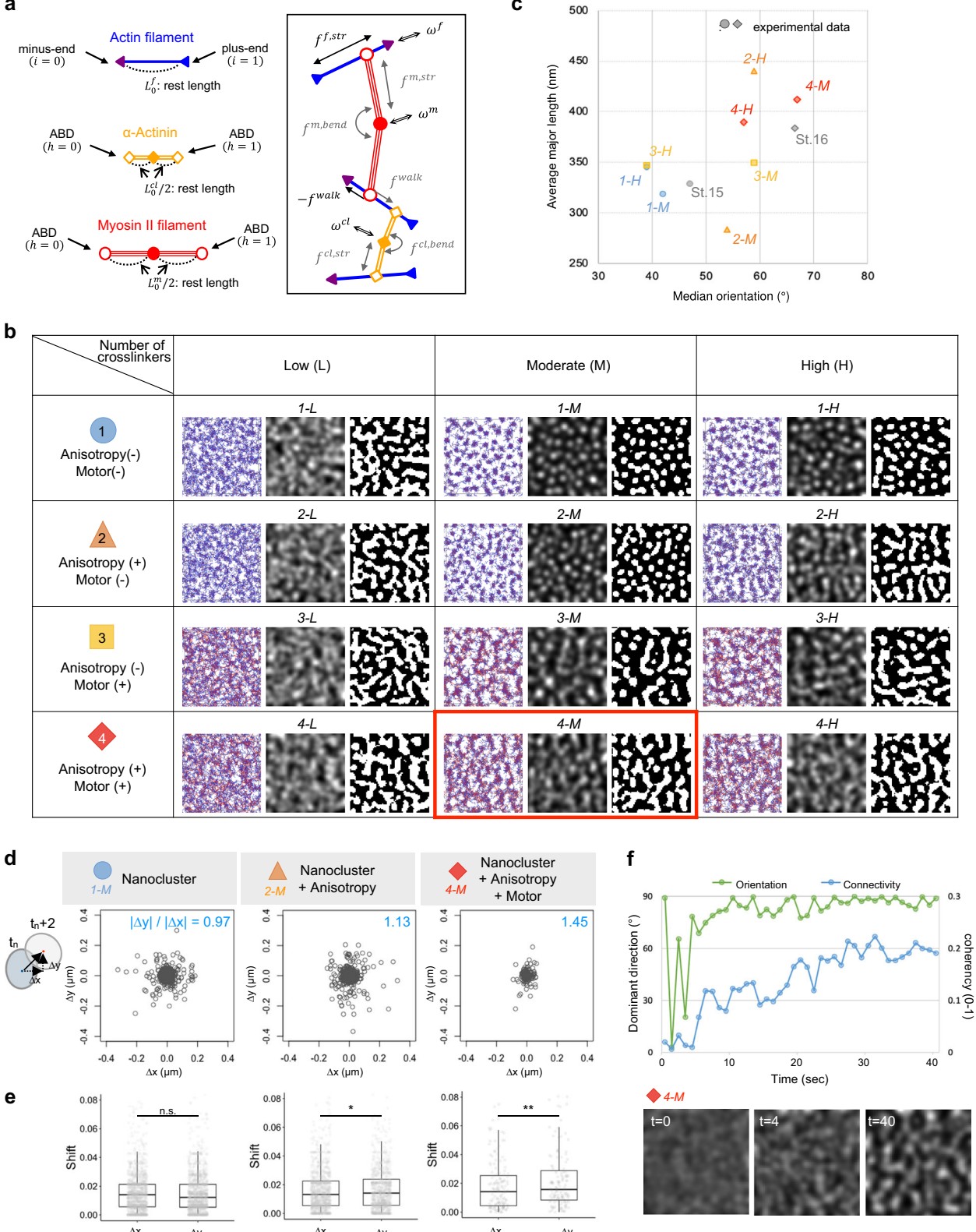

F-actin polymerization by formin mDia1[31]. Also, the pulling force applied to actin filament causes the increase of the directional polymerization of actin filament[32–34]. These mechanosensitive functions of formin proteins in the cell cortex make DAAM a prime candidate responsible for anisotropic membrane tension response in tracheal cells. Altogether, the consistency of the experimental and simulation results strongly suggests the validity of the molecular machinery proposed in this study.

The process of circumferential cable formation found in this study is reminiscent of the organization of contractile ring in fission yeast[42]. The contractile ring originates from a cytokinesis node that includes formin Cdc12, α-Actinin, and myosin II[43,44]. The Cdc12 forms the base of the node and extends F-actin filaments in a random direction[45,46]. Cdc12 also anchor Myosin II which in turn induces the circumferential coalescence of the nodes[45]. The pulling force of the myosin propagates through the actin filament resulting in Cdc12 inhibition that facilitates

**Fig. 4 | Self-organized structures reproduced by agent-based model simulation.** **a** Schematics of the simulation model, including actin filaments (blue) and two types of connectors: crosslinkers (yellow) and motors (red). See Supplementary Notes for details. **b** Steady-state solutions for different conditions. Each column refers to different numbers of cross-linkers: Low (L), moderate (M), and High (H) from left to right. Each of the four rows refers to no anisotropy and no motor (blue circle, "1"); with anisotropy and no motor (orange triangle, "2"); no anisotropy with motors (yellow square, "3"); and with anisotropy and motors (red diamond, "4"). See Supplementary Table 3 for detail information. In each condition, the snapshot (left), the filament density distribution (middle), and the binarized filament density distribution (right) are displayed. The actin pattern that meets the criterion of circumferential cable formation is marked by a red frame (*4-M*). **c** Characterization of the clusters. The median orientation and average major length are quantified using the binarized filament density distribution ("OrientationJ analysis" and

"Cluster analysis"). Each data point corresponds to the simulation condition in (**b**). For comparison, the results of the experiments at stages 15 and 16 are also plotted (gray). **d**, **e** Spatial displacement of actin clusters in x and y directions, Δx (μm) and Δy (μm), respectively. The mean value of |Δy| divided by mean value of |Δx| is displayed in blue in (**d**), and the magnitude is statistically analyzed with two-sided Student's *t* test in (**e**). $P = 0.511$ in *1-M* ($n = 1461$), $P = 0.0485$ in *2-M* ($n = 1134$), or $P = 0.00714$ in *4-M* ($n = 152$). *n* indicates number of clusters' motion appeared in a simulation. $^*P < 0.05$, $^{**}P < 0.01$. Boxplots represent median plus minima and maxima with lower and upper quantiles. **f** Formation of actin cables for the simulation condition (*4-M*) in (**b**). The actin pattern starting from a uniform distribution was characterized by the dominant direction and connectivity ("OrientationJ analysis"). The filament density distributions at $t = 0, 4, 40$ s are displayed at the bottom. Source data are provided as a Source Data file.

the effective node coalescence[47]. The mechanosensitive functions of formin well explain the anisotropic fusion of nanoclusters: the circumferentially biased F-actin polymerization by positive regulation to the local tensile stress, which will lead to the higher chance of myosin-dependent coalescence along the circumferential direction. Our study suggested that the formin might generally function in the anisotropic actin cable formation. To address this possibility, imaging with higher spatiotemporal resolution is required in future studies.

In contrast to the cytokinetic nodes that eventually converge into a single cable and create a strong force, the tracheal nanoclusters form regularly spaced multiple cables to generate a uniformly distributed circumferential contractile force, which would restrict lumen expansion throughout the long axis of the tracheal tubule[14]. The convergence of cytokinetic nodes depends on RhoA activation at the central spindle induced by the recruitment of the centralspindlin complex and the RhoGEF Ect2 to the division plane[48–50]. In tracheal cells, the distributed RhoGEF2 proteins throughout the apical cortex (Supplementary Fig. 5c) may locally activate RhoA, thus resulting in the formation of multiple nanoclusters and cables. The eventual pattern is adjustable by the regulation of the size and interval of clusters through changing turnover rates of actin filaments and crosslinkers (Supplementary Tables 3 and 5). Therefore, in addition to the formin-dependent anisotropic response, the control of molecular turnover rates causes a drastic difference in the way of actomyosin force generation.

F-actin-rich clusters that undergo rapid assembly into large condensates and abrupt disassembly have been observed in many biological contexts[23]. Actin nanoclusters in the tracheal cells associated with expanding apical membranes remained as small clusters; this was due to the expansion of the chitin matrix in the lumen. In the chitin-deficient mutant, the cortical actin was ectopically assembled into a large condensate that underwent dynamic instability[8–10]. The membrane expansion force may suppress the dynamic instability of the actin nanoclusters, allowing steady-state movement and fusion.

The actin self-organization mechanism was investigated by using an agent-based model. In the simulation, the periodic cable-like structures appeared under the existence of a sufficient number of motors in addition to the filaments, crosslinkers, and friction anisotropy. The simulation result (Fig. 4b, *4-M*) partially reproduced the clear periodic cable pattern seen in tracheal cells at early stage 16 (Fig. 1d). To improve the model, we might need to implement other factors for instance a feedback mechanism between actin cables and extracellular matrix to stabilize the cable structure, which has been previously reported[11].

Cell junctions was suggested as a source of tube cell polarity information[13,18]. Longitudinal and circumferential cell junctions differ in the tensile state of the tubules with high luminal pressure, and the Crumbs apical protein required for apical membrane growth is enriched in the longitudinal cell junction[51]. RhoGEF2 was demonstrated to be transiently enriched in the longitudinal cell junction of the larval

trachea and proposed to be an upstream signal of actin cable orientation[18]. However, we found no anisotropy in the junctional RhoGEF2 concentration in the embryonic trachea. We propose that actin nanoclusters explore the cortical membrane field, and their sensitivity to tension anisotropy enables the assembly of parallel actin cables. This emergent bottom-up orientation sensing model can be applied to a broader context of tissues or cells with no cell junctions, such as the *C. elegans* hypoderm[3] and mitotic cells[42–50].

## Methods

### *Drosophila* genetics and husbandry
All fly stocks were obtained from the Bloomington *Drosophila* Stock Center (BDSC) unless otherwise specified. The flies were maintained on standard cornmeal-yeast food at 25 °C. The *UAS-lifeact::EGFP* and *UAS-lifeact::mScarletx2* transgenic flies were generated using standard phiC31 transgenesis onto an attP40 site or attP2 site. Experiments were performed using both sexes since the sex difference in the tracheal pattern in embryonic stages 15 and 16 is not reported. The genotypes of all fly lines used in this study are listed in Supplementary Table 6.

### Molecular cloning
The pUASTattB-lifeact::mScarletx2 was constructed as follows: A DNA fragment was amplified with KOD plus Neo (TOYOBO, Japan) using pmScarlet_C1 (Addgene, #85042) as the template, with the following primer pairs: For_SS#38 "AGGGGGATCCACCGGTCGCCACCATGGT GAGCAAGGGCGAGGC" and Rev_SS#39 "GCCGCGGCCGCAGATCTA CTTGTACAGCTCGTCCATGCCG". The pUASTattB-lifeact::GFP11x3 plasmid[52] was digested at *Age*I and *Bgl*II sites then the fragment was fused using In-Fusion HD cloning Kit (Clontech), resulting in an intermediate construct pUASTattB-lifeact::mScarlet. Another DNA fragment was amplified using the same template with the primer pairs: For_SS#38 (same as above) and Rev_SS#40 "CATGGTGGCGACCG GACCTCCGCCCTTGTACAGCTCGTCCATGCCG". The pUASTattB-lifeact::mScarlet was digested by *Age*I site then the second fragment was fused to produce pUASTattB-lifeact::mScarletx2. The constructs were confirmed by sequencing.

### Immunohistochemistry
The following dyes, antibodies, and dilutions were used: Alexa Fluor 488 Phalloidin (ThermoFisher, A12379, 1:100), rabbit anti-GFP (Medical & Biological Laboratories Co., Ltd, #598, 1:500). For the fixation, freshly opened 16% Formaldehyde (Pierce, #28906) was diluted to 4% by cytoskeleton-preserving buffer (PEM: 80 mM PIPES pH 6.8, 5 mM EGTA, 2 mM MgCl₂)[53].

### Sample preparation
Embryos collected on apple agar overnight at 25 °C were dechorionated with commercial bleach and were fixed by mixing vigorously in the mixture of 4 ml 4% formaldehyde in PEM and 4 ml heptane in a liquid scintillation vial for 40 min at room temperature. After removing

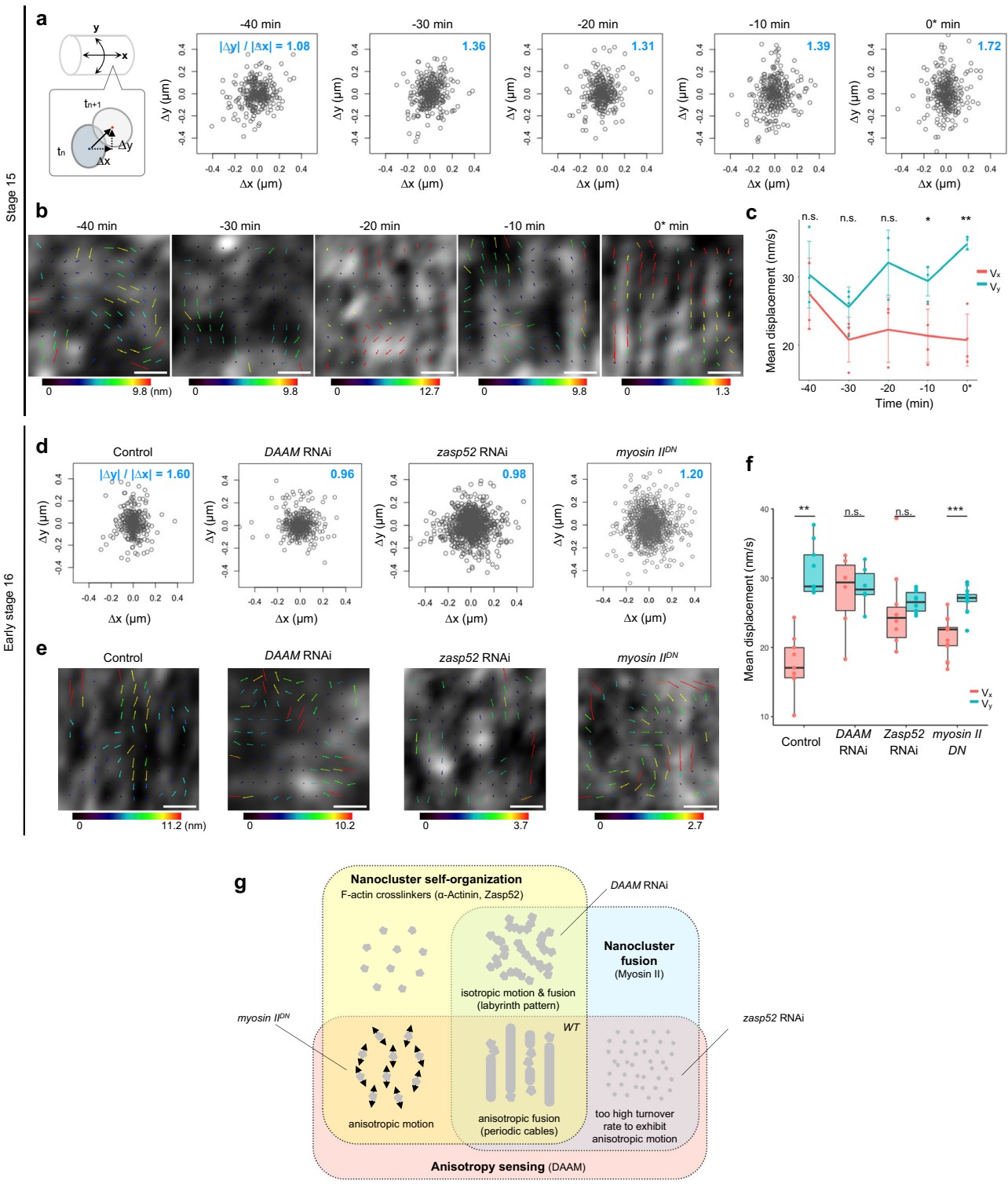

heptane, embryos were devitellinized by adding 4 ml methanol and vigorous mixing for 45 s. Devitellinized embryos were washed two times with methanol and rehydrated with PBS. Embryos were stained as described previously[54]. Blocking and antibody incubation was performed with 1% BSA, 0.2% Triton X-100 and 0.2% Tween in PBS (BW hereafter). For the phalloidin staining, the dechorionated embryos were fixed by mixing vigorously in the mixture of 4 ml 4% formaldehyde/PEM and 4 ml heptane for 10 min at room temperature, then heptane was removed. The additional fixation with 8 ml of 4% formaldehyde/PEM was performed for 30 min at room temperature, washed with BW then hand-devitellinized. The phalloidin staining was

done at 4 °C overnight. To perform live imaging, embryos were dechorionated with bleach and mounted on a glass bottom dish. Water was gently added to the dish, subsequently.

## Fluorescence microscopy
Images were captured using an inverted Zeiss LSM880 or LSM980 confocal microscope equipped with an Airyscan detector, piezo stage, and a Plan-Apochromat 63x/1.4NA oil immersion objective lens. The super-resolution images were reconstituted by the 3D auto setting of the Zen software (Carl Zeiss). For the sub-second live imaging, fast acquisition modes (by using Fast module with LSM880, or Multiplex4Y

**Fig. 5 | Circumferential motion of the actin nanoclusters is DAAM- and Zasp52-dependent. a** The Motion anisotropy analysis of small actin nanoclusters during circumferential cable formation at stage 15. The $x$ axis corresponds to the longitudinal axis, whereas the $y$ axis corresponds to the circumferential axis of the tracheal tube. The circumferentially biased motion indicated by the ratio ($|\Delta y|/|\Delta x|$) larger than unity became apparent from −30 min. **b** Particle image velocimetry (PIV) analysis of the cortical actin dynamics in tracheal cells. The velocity vectors color-coded with the vector length are superposed. The minimum and maximum lengths (nm) of the vectors are indicated below the color box. Scale bar, 0.5 μm. **c** Transition of mean displacement (nm/s) of the nanoclusters from isotropic phase to anisotropic phase calculated by the velocity vectors of PIV analyses. Line plots represent mean ± SD. The two-sided Student's $t$ test was performed for the statistical analysis. $P = 0.466$ in −40 min ($n = 4$), $P = 0.0729$ in −30 min ($n = 4$), $P = 0.0657$ in −20 min ($n = 3$), $P = 0.0176$ in −10 min ($n = 4$), or $P = 0.0038$ in 0* min

($n = 4$). *$P < 0.05$, **$P < 0.01$. **d** The Motion anisotropy analysis of small actin nanoclusters in the genetically manipulated tracheal cells at early stage 16. **e** The PIV analysis of the cortical actin dynamics. Scale bar, 0.5 μm. **f** Comparison of mean displacement (nm/s) calculated via PIV analyses. Boxplots represent median plus minima and maxima with lower and upper quantiles. The two-sided Student's $t$ test was performed for the statistical analysis. $P = 1.78e-6$ in Control ($n = 9$), $P = 0.728$ in *DAAM* RNAi ($n = 6$), $P = 0.436$ in *zasp52* RNAi ($n = 10$), or $P = 8.98e-6$ in *myosin II*$^{DN}$ ($n = 13$). *$P < 0.05$, **$P < 0.01$. **g** Venn diagram showing the relationship between three factors that regulate actin patterning in silico: Nanocluster self-organization enabled by the interaction of actin filaments and crosslinkers (light yellow); Nanocluster fusion induced by myosin II contractility (light blue); and Anisotropy sensing (light red). The phenotypes of RNAi for nanocluster components are well categorized in the Venn diagram. See "Discussion" for details. Source data are provided as a Source Data file.

setting in LSM980) were used. The $z$ sections (0.3 μm × 3) with ×10 optical zoom were scanned for 200 frames every 0.32 s. The maximal intensity projection of Z-stacks was processed using Fiji to adjust the brightness and contrast. To ensure the quality of the images, we used images whose ratio of the maximum intensity divided by minimum intensity was bigger than three in the first frame. The auto bleach correction (exponential fit) was applied for the sub-second live imaging. The room temperature was kept around 25 °C while imaging.

### Image analyses and statistical analysis
For image analysis of the actin patterns, the following methods are applied:

**OrientationJ analysis.** For dominant direction, coherency, and median orientation: An ImageJ plug-in OrientationJ[55] (http://bigwww.epfl.ch/demo/orientation/) was used. The dominant direction and coherency were used to track the transition of the actin pattern in live images, whereas the median of orientation distribution was used to accurately evaluate the orientation of fixed images. The dominant direction of an image is defined based on the gradient structure tensor of the raw image, then the degree of alignment of the patterns toward the dominant direction is defined as the coherency. A coherency close to 1 indicates a strongly aligned orientation of the patterns in the dominant direction, whereas a coherency close to 0 denotes no preferential orientation. In this study, we found the coherency was less increased by the circumferential orientation of each nanocluster but was definitely increased by circumferential fusion, consistent with the skeleton analysis. Thus, coherency was considered as an indicator of the connectivity of nanoclusters. Regions of interest (ROIs) of 50 × 50 pixels were manually selected from the medial region of the apical membrane and then applied to the "Dominant direction". If the direction was greater than 75° and the coherency was above 0.3, the actin pattern was defined as "circumferential cables formed", and the earliest time point that reached the criteria was set to 0* min. For quantification of the median of orientation distribution of RNAi screen positives and simulation results, the ROIs (75 × 75 pixels for tracheal cells and 69 × 69 pixels for simulation results) were applied to the "Direction distribution" using default settings. The average distribution of more than 18 ROIs from more than 6 embryos was calculated. The average distribution of the angles ranging from −90 to 90° was converted into axial orientation, 0–90, then the median orientation was analyzed.

**Cluster analysis.** For major length, minor length, AR, and angle of major axis: For the fitting of actin clusters to ellipses, the background of ROIs (50 × 50 pixels$^2$) was subtracted with a rolling ball ($r = 50$ pixels), binarized with Auto Local Threshold with the Otsu method ($r = 5$ pixels), then Despeckled. The binarized image was processed "Analyze particle" with a size range between 0.01 μm$^2$ to infinity, and measured the area size, major length, minor length, angle of major axis,

circularity, and solidity. After excluding clusters with a solidity of less than 0.7, the distributions of the parameters were plotted using R (https://www.r-project.org/). The angles of the major axis ranging from 0 to 180° were converted into 0 to 90, and then the mean and median orientation were analyzed by R circular statistics.

**Fusion stability.** After the binarization as written above, the constricted regions of the actin clusters were searched by the watershed segmentation, then those regions with more than 4 pixels were listed as "potential fusion sites" (#1 - #4 at $t_n$, Supplementary Fig. 2a). Then the actin signal of a single frame before ($t_{n-1}$) at the boundary was checked. If the actin signal did not exist at $t_{n-1}$, the boundary is defined as newly formed between $t_{n-1}$ and $t_n$ due to the fusion of adjacent clusters (#1 in Supplementary Fig. 2a). The newly fused site was tracked for the subsequent 11 frames to count a number of frames that more that 80% of actin signals were maintained (Supplementary Fig. 2b). The boundaries were divided into three groups based on its orientation (0–30, 30–60, 60–90°), then their durations were plotted using R.

**Motion anisotropy.** The motility of the small actin nanocluster was measured using an original program written in Matlab (https://github.com/SayakaSekine/Nanocluster_motion). Briefly, after the binarization of the images, an actin cluster within the size range 0.01 to 0.1 μm$^2$ and circularity 0.5–1.0 was selected at $t_n$, then in the next frame ($t_{n+1}$), an actin cluster with a similar size (between 0.5 to 2 fold of the size at $t_n$) and similar position (at least overlapped 1 pixel with the cluster at $t_n$) was assumed as the same cluster which moved during the single time frame. By comparing the coordinates of the center of masses between $t_n$ and $t_{n+1}$, the displacement $\Delta x$ and $\Delta y$ were calculated. The distribution of displacement was plotted, and the average ratio of absolute values ($|\Delta y|/|\Delta x|$) was calculated by R.

**PIV analysis.** For particle velocimetry image (PIV) analysis, the ROIs of 50 × 50 pixels$^2$ from the time-lapse images taken with a 0.32-s interval for 100 frames were 10 times enlarged, and the time interval was also subdivided into four using a plug-in, ZT interpolation. Then the images were applied to a plug-in, TPIV, with a window size set to 91 pixels (corresponding to 0.394 μm), and the Length ratio limit set to 0.1. The velocity vectors of each window were collected, and the average $XY$ components, $|V_x|$ and $|V_y|$, were calculated. The $x$ axis indicates the longitudinal axis, whereas the $y$ axis indicates the circumferential axis.

**Other analysis.** The actin skeleton lengths were measured after binarization as written above. Subsequently, the skeleton was analyzed with the option of prune ends. After accumulating the results from more than 5 ROIs, the average of the maximum skeleton lengths was plotted using R. The diameter of the tracheal tube was measured in the middle of TC5 and DB6, and the length of the tube has measured as a distance between the center of the lumen at TC5 and DB6 using the homemade ImageJ plug-in, CorssSectionViewer. The interval between

circumferential actin cables was measured manually. The junctional RFP::RhoGEF2 signals were extracted by using a freehand line. The mean intensity was compared between the longitudinal junction and circumferential junction, which was determined by the orientation of the junction. The degree of colocalization between the protein of interest and actin (labeled by lifeact::GFP or lifeact::mScarletx2) was evaluated as Pearson's R coefficient (above threshold) by Coloc2 in Fiji.

## Statistics and reproducibility

The statistical tests were performed using R (https://www.r-project.org/) to obtain the p values. Asterisks indicate statistical significance ($*P < 0.05$, $**P < 0.01$, and $***P < 0.001$). No statistical method was used to predetermine the sample size. To ensure the quality of the images, we excluded images whose ratio of the maximum intensity divided by minimum intensity was smaller than 3 in the first time frame. Otherwise, no data were excluded from the analyses. The experiments were not randomized. The investigators were not blinded to allocation during experiments and outcome assessment.

## RNAi screens

For the first RNAi screen, the transgenic RNAi males were crossed in individual vials to *btl-gal4* virgin females (25 flies/vial) at 25 °C. One day later, flies were flipped to new vials, and progeny were kept at 25 °C for 13 days to score pupation timing. The number of pupae was counted on 5, 6, 7, and 8 days after the crossing. If the number of pupae was less than 70 on day 8, the RNAi line was judged as "pupation defective", and used in the second screen (Supplementary Table 2). For the second screen, the transgenic RNAi males were crossed in individual vials to *btl-gal4 UAS-lifeact::GFP* virgin females at 25 °C. The embryos were collected and fixed as written above, then the microscopic observation was performed.

## Molecular interaction network

The molecular interaction network was drawn by using the Molecular Interaction Search Tool (MIST, ver 5.0, https://fgrtools.hms.harvard.edu/MIST/). The input genes were as follows: *CG3630, ZaspS2, Chd64, tsr, ssh, cora, dia, form3, DAAM*, and *wupA* (10 genes). The settings for filtering are written in Supplementary Fig. 4. The network output by the MIST was organized manually to align all input genes (screen positives) placed on an outer circle, and the hub genes (connecting more than 3 genes) were placed in the center (blue circle is manually added). The molecular complexes found in the network are listed in Supplementary Table 4.

## Models

The coarse-grained molecular dynamics simulation model details are described in the Supplementary Notes. The reconstruction of fluorescent signal-like pseudo images for analyses was prepared in the following manner. To calculate the filament density distribution, the simulation box ($3 \times 3\,\mu m^2$) was discretized into $69 \times 69$ square boxes of the length $0.04325\,\mu m$, which is comparable to the pixel size of the experimental images. Then, the number of filament particles in each box was counted. The number of actin filaments in each pixel was calculated, and the value was multiplied by 20 to match the intensity value of experimental images. Then the image was processed with a Gaussian blur filter (sigma = 2) to mimic the diffusion of light.

## Structure factor

The structure factor is calculated from the position of the filament particles as

$$S(q) = \int_{q}^{q+\delta q} \int_{0}^{2\pi} \left\langle \frac{1}{N_f} \sum_{j} \sum_{k} \exp\left[-i\,\boldsymbol{q} \cdot \left(\boldsymbol{r}_j^f - \boldsymbol{r}_k^f\right)\right] \right\rangle q\, d\theta\, dq$$

where each summation runs over all filament particles and the integral $\int_{0}^{2\pi} d\theta$ is calculate over the angle of the $\boldsymbol{q}$ vector. The integral $\int_{q}^{q+\delta q} dq$ is calculated to separate the data into bins with the width $\delta q$. $\langle \cdot \rangle$ represents the time average. The interval of the structure is measured by fitting the data $S(q)$ around the peak by using the Gaussian function $\frac{A}{(2\pi)^{1/2}\sigma} \exp[-\frac{(q-\mu)^2}{2\sigma^2}]$ as $\frac{2\pi}{\mu}$.

## Reporting summary

Further information on research design is available in the Nature Portfolio Reporting Summary linked to this article.

## Data availability

The image data generated and/or analyzed in this study have been deposited in the SSBD:repository database under accession code https://doi.org/10.24631/ssbd.repos.2023.12.330. Source data are provided with this paper.

## Code availability

All homemade ImageJ plugins and Matlab codes are available at https://signaling.riken.jp/tools/imagej-plugins/. The Matlab code for motion anisotropy is also available at https://github.com/SayakaSekine/Nanocluster_motion. The simulation code is available upon request to M.T.

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

## Acknowledgements

We thank Dr. K. Röper, Dr. F. Schöck, Dr. M. Llimargas, Dr. J. Grosshans, Bloomington, and Kyoto Drosophila Stock Centers for fly strains. We thank N. Morimitsu for technical support. We also thank Y.-C. Wang and Y. Uchida for helpful comments; and members of S. Hayashi, Y.-C. Wang, T. Nishimura, S. Yoo, and E. Kuranaga laboratories for discussions. This work was supported by JSPS Grant-in-Aid for JSPS fellows (18J40165 to

S.S.), JSPS Grant-in-Aid for Early-Career Scientists (18K14746 to S.S. and 19K14673 and 22K14017 to M.T.), JSPS Grant-in-Aid for Scientific Research (22K06214 to S.S. and M.T., 22H05170 for T.S. and 19H00996 for S.H. and T.S. and 19H00996 to S.H.), Otsuka Pharmaceutical-RIKEN CDB COCC program (to S.S. and S.H.) and Naito Foundation Research Grant for Woman Scientist (to S.S.).

## Author contributions

S.S. and S.H. conceived the study and designed experiments. S.S. performed the experiments. M.T. performed the simulations. H.W. and M.M.S. wrote codes for image analysis, and S.S. analyzed the data. S.S. and M.T. wrote the first version of the paper, and T.S. and S.H. contributed to the interpretation and discussion of the data. All authors reviewed the manuscript.

## Competing interests

The authors declare no competing interests.
