## [Peer Review File · Nature Communications]

Emergence of periodic circumferential actin cables from the anisotropic fusion of actin nanoclusters during tubulogenesisREVIEWER COMMENTS

Reviewer #1 (Remarks to the Author):

Periodically spaced circular f-actin structures wrap around the circumference of axons and epithelial tubes, where they act as scaffolds that provide mechanical support and maintain axonal or tubular shape. However, the molecular mechanisms underlying actin ring formation are not well understood. Sekine et al. used imaging, RNAi-based functional analysis, and computational modeling to investigate the formation of actin rings in tracheal tubes of *Drosophila* embryos. The authors show that cortical f-actin is initially present in clusters that orient along the circumferential axis of the tube as the tracheal lumen expands during embryonic development. Furthermore, they show that separate f-actin clusters fuse to generate elongated cables. Interestingly, fusion occurs in an anisotropic fashion preferentially along the circumferential axis of the tube. The authors propose that tension anisotropy, which is imposed by tubular geometry, leads to biased orientation and fusion of actin clusters, and that this spatially biased fusion over time yields the pattern of circumferential actin rings.

To identify factors required for the fusion and re-orientation of f-actin clusters, they carried out an RNAi-based screen of actin-binding proteins. This revealed several genes whose knockdown in tracheal cells led to defects in formation or orientation of actin rings. They focused on characterizing the roles of the actin filament crosslinker alpha-actinin, the alpha-actinin-associated protein Zasp52, and the formin DAAM. DAAM RNAi led to abnormal orientation of actin clusters, but clusters still fused, whereas depletion of alpha-actinin, Zasp52, or RhoA disturbed cluster fusion, suggesting that distinct mechanisms control orientation and elongation of annular actin cables. In the rest of the paper, the authors describe a coarse-grained molecular dynamics model to simulate self-organization, anisotropic orientation, and fusion of actin clusters. I cannot judge the mathematical basis and the validity of results of the simulations. Although the model seems to make interesting predictions about the roles of actin crosslinkers, formins, and myosin, the authors do not go further to test these predictions experimentally. However, this would be crucial to validate some of the major conclusions, such as the idea that myosin II mediates actin cluster fusion (see comments below).

Overall, the paper describes interesting observations derived from a careful quantitative analysis of actin ring formation, and thereby contributes to a better understanding of this important biological process. However, the work would clearly benefit from better integrating the modeling approach with experimental data, and from making the modeling part more accessible to the non-expert reader.

The authors would need to address the following points:

The authors attribute distinct roles, such as "tensile stress sensing" to specific molecules (alpha-actinin, DAAM, myosin II, Zasp52). While a requirement in actin ring formation is supported by RNAi experiments, precise roles of these proteins are predicted by theoretical simulations that are not experimentally tested. Given this concern, some of the major conclusions appear clearly too strong or

premature. For instance, I see no direct evidence that DAAM is "sensing" tensile stress or anisotropy, as stated in abstract and main text. Based on experimental evidence presented in the paper, DAAM could equally well mediate a response to anisotropic stress, rather than sensing it. The authors need to carefully rephrase this and related statements throughout the text.

The abstract claims a role of myosin 2 in "nanocluster fusion", but aside from colocalization and modeling data, no functional evidence is presented to substantiate this idea.

Line 176: "Myosin did promote the fusion of clusters (Fig. 4b,c, 3-M, 3-H, Supplementary Table 5)": what is "3-M, 3-H" referring to? I cannot find these image panels.

Line 28: "simulation demonstrates that crosslinkers play a crucial role": "demonstrate" is an overstatement, simulations can at best suggest a role. Please rephrase.

The authors state that they used a pixel reassignment technique that enables super-resolution imaging, but in Materials and Methods they describe what appears to be regular confocal/AiryScan microscopy. As a reference, they cite a review about different super-resolution techniques. Please explain and cite a proper reference for what was used.

I have difficulties seeing nanocluster fusion events on the images and in the movies. In Fig. 2a, an example is shown for fission (though it is not clear what "cable #2" means). Please clarify and highlight an example of cluster coalescence/fusion.

Fig. 1c,d: I see circumferentially oriented actin cables already at stage 15, although the text states that they appear only by stage 16. Please explain.

I have difficulties to see "strong colocalization" (as stated in line 129) for DAAM and alpha-Actinin in the image shown in Supplemental Fig. 5C.

The resolution of the videos appears rather low, can this be improved by lower compression of the movie files?

Line 83: "connectivity and orientation were initially low": For orientation (Fig. 1j), the most relevant result appears to me that the variation in orientation decreases substantially from very high variation at $t=-20$ min to essentially no variation by stage 16 ($t=0$ min), indicating that all nanoclusters are aligned in the same orientation by that stage.

Line 140: The authors argue that the increased duration of fused states of clusters promotes the formation of circumferentially oriented actin cables. However, although the "duration of circumferential fusion was significantly longer than that of diagonal or longitudinal fusion in control and *zasp52* RNAi-expressing tracheal cells (Fig.3e)" (line 140), *zasp52* RNAi led to strong orientation defects (Fig. 3b). This appears to weaken the authors' previous argument and needs to be discussed.

Fig. 1 e-g and line 568: It is not clear to me what exactly was tested in the statistical test. The Mode value of distributions of each embryo? The mode value looks almost identical between all the graphs in e-g. Probably a distribution test like Kolomogorov-Smirnov would be more appropriate here.

Minor comments:

Fig. 1h: Axis labels are missing.

Fig. 1j-m: What do the vertical lines at $t=-60$ min and $t=0$ min indicate? Please explain in Figure legend.

Line 58: "we examined the emergent processes of periodic actin cables". What does this mean?

Line 104, Fig. 2b: "... became more significant at 0 min": Delete "more".

line 240: Replace "crumb" by "Crumbs".

Line 36: Biological systems are commonly not considered the result of "rational design". Please rephrase.

Reviewer #2 (Remarks to the Author):

This study investigates the formation of a circumferential actin cable in a *Drosophila* tracheal tube. Researchers discovered that actin nanoclusters, which sense stress in the membrane, exhibit biased motion and fusion, eventually forming a larger cable structure. They claimed that the proteins DAAM

and non-muscle myosin II play crucial roles in this process, whereas crosslinkers and mechanical anisotropy also contribute. Although the experimental observations seem to be interesting, there are many issues in this manuscript as commented below.

- The mergence of nodes into a linear structure observed in this study is reminiscent of how the cytokinetic ring is formed from nodes in fission yeasts (DOI: 10.1007/s12551-018-0476-6). Although the authors briefly mentioned the relevance of this finding to mitotic cells at the end of the main text, the discussion should be more extensive. Indeed, there are many similarities between two processes in terms of key proteins (alpha-actinin and myosin).

- The analysis of clusters is based on binarized images, which can exaggerate the size of clusters as can be seen in Fig. 1c. I don't understand why the authors didn't use super-resolution imaging technique to find the real size of clusters rather than using binarized images. Since conclusions and the modeling setup depend on the cluster size observed in experiments, the use of binarized images is highly concerning part in this study.

- The authors mentioned that "Nanocluster fusion in the circumferential direction was preferentially stabilized; thus, the actin nanoclusters served as building blocks for cable formation". However, from the images, it's unclear whether the fusion of nanoclusters is stabilizing. In some cases, it may be stabilizing. However, as the visualization is contradictory, thus it is difficult to make a statement. For example, In Fig. 2a, cables #3 & 4 of the 0-min series, the cables are breaking down to nanoclusters and then reappear once again, which makes it highly unstable. Maybe the authors could tone down the statement or give some more proof of the stabilization of the fusion process.

- The authors said "the simulation demonstrated that the elliptical actin nanoclusters react to friction anisotropy and elongate in the direction of higher friction, developing into regular cables." Although simulations show the biased orientation of the nanoclusters in the red case (+anisotropy + motor), I don't see any structure close to Fig. 1d in simulations results (Fig. 4b). The authors should mention the limit of insights from their model.

- The architecture of F-actins and cross-linkers in each nanocluster is not clear at all based on experimental results. Nevertheless, the model used arbitrary assumptions and reached a conclusion "Further analyses revealed that the clusters were formed through the competition between the effective diffusion of the filaments and their effective attraction due to the crosslinkers connecting the filaments." How do you know whether the nanoclusters formed in the model will share the same origin with those in experiments?

- In their model (Fig. 4b), the authors found that the crosslinking density significantly affects cluster formation, and including the motor in the network enhances fusion, making the cable-like structure. However, it's still unclear whether motor density has any effect on producing cable-like structures.

- The number of data points in Fig. 1 (j-k) is insufficient to observe the trend clearly. Increasing the number of data points would result in a smoother curve and improve the visualization.

- In Fig. 2b and 2c, there is a mismatch between the color code and the mentioned direction. The blue color (0 deg-30 deg) should represent longitudinal, but in 2c, it is shown as circumferential. The figures should be made consistent to avoid confusion.

- The type of simulations used in this study is not (coarse-grained) molecular dynamics simulations. It is more like an agent-based model or a discrete model.

Response to reviewer comments

Reviewer #1 (Remarks to the Author):

Periodically spaced circular f-actin structures wrap around the circumference of axons and epithelial tubes, where they act as scaffolds that provide mechanical support and maintain axonal or tubular shape. However, the molecular mechanisms underlying actin ring formation are not well understood. Sekine et al. used imaging, RNAi-based functional analysis, and computational modeling to investigate the formation of actin rings in tracheal tubes of *Drosophila* embryos. The authors show that cortical f-actin is initially present in clusters that orient along the circumferential axis of the tube as the tracheal lumen expands during embryonic development. Furthermore, they show that separate f-actin clusters fuse to generate elongated cables. Interestingly, fusion occurs in an anisotropic fashion preferentially along the circumferential axis of the tube. The authors propose that tension anisotropy, which is imposed by tubular geometry, leads to biased orientation and fusion of actin clusters, and that this spatially biased fusion over time yields the pattern of circumferential actin rings.

To identify factors required for the fusion and re-orientation of f-actin clusters, they carried out an RNAi-based screen of actin-binding proteins. This revealed several genes whose knockdown in tracheal cells led to defects in formation or orientation of actin rings. They focused on characterizing the roles of the actin filament crosslinker alpha-actinin, the alpha-actinin-associated protein Zasp52, and the formin DAAM. DAAM RNAi led to abnormal orientation of actin clusters, but clusters still fused, whereas depletion of alpha-actinin, Zasp52, or RhoA disturbed cluster fusion, suggesting that distinct mechanisms control orientation and elongation of annular actin cables. In the rest of the paper, the authors describe a coarse-grained molecular dynamics model to simulate self-organization, anisotropic orientation, and fusion of actin clusters. I cannot judge the mathematical basis and the validity of results of the simulations. Although the model seems to make interesting predictions about the roles of actin crosslinkers, formins, and myosin, the authors do not go further to test these predictions experimentally. However, this would be crucial to validate some of the major conclusions, such as the idea that myosin II mediates actin cluster fusion (see comments below).

Overall, the paper describes interesting observations derived from a careful quantitative

analysis of actin ring formation, and thereby contributes to a better understanding of this important biological process. However, the work would clearly benefit from better integrating the modeling approach with experimental data, and from making the modeling part more accessible to the non-expert reader.

The authors would need to address the following points:

The authors attribute distinct roles, such as "tensile stress sensing" to specific molecules (alpha-actinin, DAAM, myosin II, Zasp52). While a requirement in actin ring formation is supported by RNAi experiments, precise roles of these proteins are predicted by theoretical simulations that are not experimentally tested. Given this concern, some of the major conclusions appear clearly too strong or premature. For instance, I see no direct evidence that DAAM is "sensing" tensile stress or anisotropy, as stated in abstract and main text. Based on experimental evidence presented in the paper, DAAM could equally well mediate a response to anisotropic stress, rather than sensing it. The authors need to carefully rephrase this and related statements throughout the text.

Thank you for the important suggestion. DAAM RNAi exhibited strong phenotype in asymmetric fusion stabilization (Fig. 3e), biased motion (Fig. 5d-f) and the labyrinth pattern formation (Fig. 3f). Those phenotypes are distinct from the zasp52 RNAi which showed general fusion failure (Fig. 3b,f) and biased motion defects (Fig. 5d-f). The results suggest that DAAM is a central component of the machinery responding to the membrane tension asymmetry. By considering previous studies reporting tensile stress sensing by formin family protein (Kozlov et al., JCB, 2004, Je'gou et al., Nat Commun, 2013, Higashida et al., Nat Cell Biol, 2013, Yu et al., Nat Commun, 2017), we speculated that DAAM is a strong candidate for the asymmetric tensile stress sensor. However, we agree with this reviewer that this is an overstatement without direct manipulation of the tensile stress *in vivo*. We rephrased the manuscripts according to the suggestion as written below:

Line 27: "RNAi screening revealed the formin family protein, DAAM, as an essential component **responding to tissue anisotropy**, and non-muscle myosin II as a component required for nanocluster fusion."

Line 31: "Altogether, we propose that an actin nanocluster is an organizational unit that **responds to** stress in the cortical membrane and builds a higher-order cable structure."

Line 166: “To identify the molecules crucial for local anisotropy response, nanocluster fusion analysis was performed on fast live imaging at early stage 16.”

The abstract claims a role of myosin 2 in "nanocluster fusion", but aside from colocalization and modeling data, no functional evidence is presented to substantiate this idea.

We appreciate reviewer 1 for giving us the chance to substantiate the claim. We conducted additional experiments by using a dominant negative form of Myosin II heavy chain (*myosin II^{DN}*, *UAS-zipper[ROD]* generated in Franke et al., *Dev Biol*, 2010). The expression of *myosin II^{DN}* resulted in a failure of cluster fusion at early Stage 16 and a weakly fused pattern at late Stage 16 (Fig 3b, f) which was consistent with the claim that myosin II is required for the cluster fusion. In addition, the fast live imaging and series of image analyses (Fusion stability, motion anisotropy, and PIV analysis) revealed that the anisotropic fusion and anisotropic motion of the actin nanoclusters were still observed even with the expression of *myosin II^{DN}* (Fig 3e, Fig 5d-f), indicating that myosin II contributes less to the anisotropy response compared to DAAM or Zasp52. This is also consistent with the simulation result in Fig 3, in which condition 2-M (crosslinker + anisotropy without motor) showed the weaker but significant anisotropic motion of the clusters (Fig. 3d, e). We explained the new results in the manuscripts and left the abstract unchanged as “non-muscle myosin II as a component required for nanocluster fusion”.

Line 168: “The duration of circumferential fusion was significantly longer than that of diagonal or longitudinal fusion in control, *zasp52* RNAi, and dominant negative form of myosin II (*myosin II^{DN}*, *zip^{ROD}* in *Drosophila*)-expressing tracheal cells (Fig. 3e).”

Line 256: “In addition, the attenuation of Myosin II by the expression of dominant negative form also failed fusion (Fig. 3f), consistent with the ROCK inhibitor treatment that caused a breakdown of the actin cables to clusters.”

Line 258: “Finally, the requirement of myosin II in the motion anisotropy was checked. In contrast to the knockdown of the other two components, the anisotropy was retained but at a lower degree in Myosin II downregulation (Fig 5d-f). Hence, in addition to DAAM, the crosslinker-dependent clustering of actin filaments contributes to the anisotropy-responding machinery, whereas myosin II plays a role in cluster fusion.”

Line 176: "Myosin did promote the fusion of clusters (Fig. 4b,c, 3-M, 3-H, Supplementary Table 5)": what is "3-M, 3-H" referring to? I cannot find these image panels.

We are sorry for the inconspicuous labeling of the images. In Fig. 4b, the number "3" refers to the condition of no anisotropy with myosin motors (indicated as a yellow square on the left side), whereas "M" refers to the moderate number of crosslinkers (indicated at the top). The "3-M" is the image second from left, third from top in Fig 4b. Likewise, the "3-H" is the image third from left, third from top. To improve visibility, we enlarged the labels and added precise explanations to the legend as below.

Legend of Fig 4b (Line 557): "Steady-state solutions for different conditions. Each column refers to different numbers of crosslinkers: Low (L), moderate (M), and High (H) from left to right. Each of the 4 rows refers to no anisotropy and no motor (blue circle, "1"); with anisotropy and no motor (orange triangle, "2"); no anisotropy with motors (yellow square, "3"); and with anisotropy and motors (red diamond, "4"). See Supplementary Table 3 for detail information."

Line 28: "simulation demonstrates that crosslinkers play a crucial role": "demonstrate" is an overstatement, simulations can at best suggest a role. Please rephrase.

Thank you for your advice. We rephrased the manuscripts as suggested.

Line 28: "An agent-based model simulation suggested that crosslinkers play a crucial role in nanocluster formation and cluster-to-cable transition occurs in response to mechanical anisotropy."

The authors state that they used a pixel reassignment technique that enables super-resolution imaging, but in Materials and Methods they describe what appears to be regular confocal/AiryScan microscopy. As a reference, they cite a review about different super-resolution techniques. Please explain and cite a proper reference for what was used.

We apologize for the confusion caused by the lack of explanation for this point. Zeiss's Airyscan uses a detector composed of 32 elements, each equivalent to a point detector. Each element of the detector array acts separately, meaning that the fluorescence is captured 32 times rather than once. By correcting for the displacement of each element from the optical axis through the pixel reassignment technique, the image can be reconstructed with a higher resolution, beyond the diffraction limit (Huff, *J. Nat Methods* 12, 2015, Sheppard et al., *Optics Letters*, 38(15), 2889-2892, 2013, Korobchevskaya et al., *Photonics*,

4(41), 2017). The process of image reconstruction was done by Zen software (Carl Zeiss) with 3D auto setting as written in the Methods' "Fluorescence microscopy" section. The originality of this microscopy technique made it difficult to explain briefly in the manuscript, thus we cited the review which introduces Airyscan as an example of the pixel reassignment technique-based super-resolution microscopy technique (Valli et al., *JBC* 297, 100791, 2021). To explain the details and benefits of Airyscan for our study more clearly, we added the following sentences written below and cited the articles above.

Line 75: "As the process is highly dynamic in deep tissue and the size of the nascent actin cables was close to the diffraction limit, we used Airyscan which enables super-resolution imaging based on deconvolution and the pixel reassignment principle."

I have difficulties seeing nanocluster fusion events on the images and in the movies. In Fig. 2a, an example is shown for fission (though it is not clear what "cable #2" means). Please clarify and highlight an example of cluster coalescence/fusion.

We are sorry for the unclear visualization. The cluster coalescence/fusion sites were marked with magenta in the fluorescent and binary images in Fig 2a. Also, the images were enlarged to show the complex pattern more clearly. The legend was rephrased as follows.

Line 513: "The fusion sites of clusters were labeled with magenta (see Supplementary Fig. 2 and *Fusion stability* in Method for definition)."

Fig. 1c,d: I see circumferentially oriented actin cables already at stage 15, although the text states that they appear only by stage 16. Please explain.

As shown in Fig 1b, Stage 15 last for about 100 min and the circumferential orientation of the actin nanoclusters gradually proceeds during Stage 15 as quantified in revised Fig 1h. In our study, embryonic Stages 15 and 16 are defined by the shape of the intestine. This criterion is a widely used developmental marker (Campos-Ortega and Hartenstein, Springer, 1997), but the developmental speed of the intestine and tracheal system slightly differs among individuals. Thus, although the two images in Fig 1c are both taken from the Stage 15 embryos, they are in different phases (i.e. early or middle phases) of the actin nanocluster orientation (and fusion) in the tracheal cells. We wrote that circumferential cable appears by Stage 16, but not "only" by Stage 16.

I have difficulties to see "strong colocalization" (as stated in line 129) for DAAM and alpha-

Actinin in the image shown in Supplemental Fig. 5C.

We rephrased the “strong colocalization” to “significant colocalization” (line 155).

The resolution of the videos appears rather low, can this be improved by lower compression of the movie files?

Since the original frame sizes of the images were small, especially for the 50 x 50 pixels² ROI images, the movies appeared as low resolution. For better visualization, we enlarged the images without interpolation and then generated the movie files (revised Supplementary movies 1-3).

Line 83: "connectivity and orientation were initially low": For orientation (Fig. 1j), the most relevant result appears to me that the variation in orientation decreases substantially from very high variation at t=-20 min to essentially no variation by stage 16 (t=0 min), indicating that all nanoclusters are aligned in the same orientation by that stage.

Thank you for your suggestion for the interpretation of the result. Regarding this point, we increased the number of data points in revised Figures 1h and i as suggested by Reviewer #2. In addition, the standard deviation was indicated with the light green region. The results were consistent with Reviewer #2's suggestion, the variation in orientation was high from t=-40 min to t=-10 min, when the connectivity was still low. Thus, we changed the interpretation as follows:

Line 102: “The orientation increased over 40 min with high variation (-40 min to 0 min in Fig. 1h; time 0 corresponds to the completion of circumferential alignment of the actin pattern, Method *Other analysis*), whereas the connectivity stayed low by -10 min (Fig. 1i). When the connectivity increased at 0 min, the variation of orientation substantially decreased (Fig. 1h, i).”

Line 140: The authors argue that the increased duration of fused states of clusters promotes the formation of circumferentially oriented actin cables. However, although the "duration of circumferential fusion was significantly longer than that of diagonal or longitudinal fusion in control and *zasp52* RNAi-expressing tracheal cells (Fig.3e)" (line 140), *zasp52* RNAi led to strong orientation defects (Fig. 3b). This appears to weaken the authors' previous argument and needs to be discussed.

We think the reason for the strong orientation defects in *zasp52* RNAi is the weakening of crosslinking of the actin filaments, which in turn enhances the diffusion of actin filaments from actin nanocluster, leading to the instability of the cable structures. Although the nanoclusters exhibit anisotropy in the duration of the fused states, if the actin filaments diffuse strongly, the fused pattern (i.e. circumferential cable pattern) becomes unstable and eventually turns into the isotropic pattern. The simulation results with a low number of crosslinkers or high turnover rate of crosslinkers (Fig 4b, 4-L, and Supplementary Fig. 6d4) are consistent with this idea and help to image the molecular interaction. We added this argument in the discussion section and modified Fig. 5g to make the difference between DAAM RNAi phenotype and *zasp52* RNAi phenotype clearer.

Line 273: “Firstly, the simulation with the low number of crosslinkers (Fig 4b, 4-L) exhibited an isotropic pattern (Supplementary Table 3), which was similar to the downregulation of the crosslinker, *zasp52*, that showed orientation defect (Fig. 3a). Although the nanocluster fusion is stabilized anisotropically in *zasp52* RNAi (Fig. 3e), the weaker crosslinking might cause a high turnover of actin filaments and thus fail to maintain the anisotropic higher-order cable structure.”

Legend of Fig. 5g (Line 593): “g, Venn diagram showing the relationship between three factors that regulate actin patterning *in silico*: Nanocluster self-organization enabled by the interaction of actin filaments and crosslinkers (light yellow); Nanocluster fusion induced by myosin II contractility (light blue); and Anisotropy sensing (light red). The phenotypes of RNAi for nanocluster components are well categorized in the Venn diagram. See *Discussion* for details.”

Fig. 1 e-g and line 568: It is not clear to me what exactly was tested in the statistical test. The Mode value of distributions of each embryo? The mode value looks almost identical between all the graphs in e-g. Probably a distribution test like Kolmogorov-Smirnov would be more appropriate here.

In previous Fig. 1 e-g, the values of major axis length, minor axis length, and aspect ratio for each actin nanocluster between Stage 15 and 16 were compared with Mann-Whitney's U test. The number of **n** and **N** in the legend indicated the number of clusters and ROIs, respectively. Thanks to your advice, we got better results with the Kolmogorov-Smirnov test to compare two populations with the same data set. The following results were implemented in the revised Fig. 1e, revised Supplementary Fig. 1 b, c and legend:

Comparison of Major length: p-value = 0.009809 (**)

Comparison of Minor length: p-value = 0.329 (n.s.)

Comparison of Aspect ratio: p-value = 0.0003169 (***)

Minor comments:

Fig. 1h: Axis labels are missing.

The labels were added as follows:

x-axis: "Longitudinal axis" with a two-sided arrow.

y-axis: "Probability"

Fig. 1j-m: What do the vertical lines at t=-60 min and t=0 min indicate? Please explain in Figure legend.

The dotted vertical lines indicated the beginning and the end time points of the orientation increase of actin pattern, which matched the time window of the tube expansion. In the revised Figs 1h-k, only the dotted vertical line at t=0 was left to make it simple. The explanations were added in the Results and Figure legend.

Line 102: "The orientation increased over 40 min with high variation (-40 min to 0 min in Fig. 1h; time 0 corresponds to the completion of circumferential alignment of the actin pattern, Method *Other analysis*)"

Line 507 (Legend of Figure 1): "The dotted vertical lines in h-k indicate when the circumferential cables are formed (0 min)."

Line 58: "we examined the emergent processes of periodic actin cables". What does this mean?

The sentence was rephrased as follows to make it more concrete.

Line 74: "To study the molecular mechanism of actin cable formation, we first observed the transition of cortical actin patterns at several embryonic stages."

Line 104, Fig. 2b: "... became more significant at 0 min": Delete "more".

The more was deleted.

line 240: Replace "crumb" by "Crumbs".

The word was replaced (line 334).

Line 36: Biological systems are commonly not considered the result of "rational design".

Please rephrase.

The sentence was rephrased to "One mechanical solution for robustly maintaining the tube diameter while allowing flexibility in tube curvature is to introduce periodic circumferential actin cables." (line 36)

Reviewer #2 (Remarks to the Author):

This study investigates the formation of a circumferential actin cable in a *Drosophila* tracheal tube. Researchers discovered that actin nanoclusters, which sense stress in the membrane, exhibit biased motion and fusion, eventually forming a larger cable structure. They claimed that the proteins DAAM and non-muscle myosin II play crucial roles in this process, whereas crosslinkers and mechanical anisotropy also contribute. Although the experimental observations seem to be interesting, there are many issues in this manuscript as commented below.

- The emergence of nodes into a linear structure observed in this study is reminiscent of how the cytokinetic ring is formed from nodes in fission yeasts (DOI: 10.1007/s12551-018-0476-6). Although the authors briefly mentioned the relevance of this finding to mitotic cells at the end of the main text, the discussion should be more extensive. Indeed, there are many similarities between two processes in terms of key proteins (alpha-actinin and myosin).

Thank you for your important advice. We added the following paragraphs to the discussion.

Line 290 - 316: "The process of circumferential cable formation found in this study is reminiscent of the organization of contractile ring in fission yeast⁴². The contractile ring originates from a cytokinesis node that includes formin Cdc12, α -Actinin and myosin II^{43,44}. The Cdc12 forms the base of the node and extends F-actin filaments in a random direction^{45,46}. Cdc12 also anchor Myosin II which in turn induces the circumferential coalescence of the nodes⁴⁵. The pulling force of the myosin propagates through the actin filament resulting in

Cdc12 inhibition that facilitates the effective node coalescence⁴⁷. If similar molecular mechanisms are occurring in tracheal cells, it would explain the circumferential bias in the fusion of nanoclusters: the circumferentially biased F-actin polymerization in each nanocluster by positive regulation to the local tensile stress, which will lead to the higher chance of myosin-dependent coalescence along the circumferential direction. The mechanosensitive formin might generally function in the anisotropic actin cable formation. To address this possibility, imaging with higher spatiotemporal resolution is required in future studies.

In contrast to the cytokinetic nodes that eventually converge into a single cable and create a strong force, the tracheal nanoclusters form regularly spaced multiple cables to generate a uniformly distributed circumferential contractile force, which would restrict lumen expansion throughout the long axis of the tracheal tubule¹⁴. The convergence of cytokinetic nodes depends on RhoA activation at the central spindle induced by the recruitment of the centralspindlin complex and the RhoGEF Ect2 to the division plane^{48–50}. In tracheal cells, the distributed RhoGEF2 proteins throughout the apical cortex (Supplementary Fig. 5c) may locally activate RhoA, thus resulting in the formation of multiple nanoclusters and cables. The eventual pattern is adjustable by the regulation of the size and interval of clusters through changing turnover rates of actin filaments and crosslinkers (Supplementary Tables 3,5). Thus, in addition to the formin-dependent anisotropic response, the control of molecular turnover rates causes a drastic difference in the way of actomyosin force generation.”

- The analysis of clusters is based on binarized images, which can exaggerate the size of clusters as can be seen in Fig. 1c. I don't understand why the authors didn't use super-resolution imaging technique to find the real size of clusters rather than using binarized images. Since conclusions and the modeling setup depend on the cluster size observed in experiments, the use of binarized images is highly concerning part in this study.

We understand your concern, and thank you for pointing out the important point. In this study, we aimed to capture the dynamic process of the actin cortex patterning at the tracheal tube *in vivo*. Because the tracheal luminal membrane is located at the inner side of the tissue, the focal planes needed to be set between 10 to 30 μm deep, meaning that super-resolution techniques such as TIRF, PALM/STORM and STED are difficult to apply to the whole mount live imaging. At the same time, since the size of the circumferential cable was close to the diffraction limit, a super-resolution imaging technique with high speed was necessary. Based on these requirements, we have decided to use Zeiss LSM with the

Airyscan detector. Airyscan uses a detector composed of 32 elements, and by correcting for the displacement of each element from the optical axis through the pixel reassignment technique, the image can be reconstructed with a super-resolution and improved signal-to-noise ratio. It successfully visualized the behavior of the actin nanoclusters with a sub-second temporal resolution, and the easy accessibility enabled the RNAi screen, leading to the conceptual findings in this study.

As Reviewer #2 suggested, the size estimation based on binarized images is less accurate and contains errors to some degree. However, to accomplish the resolution and speed of the imaging required in this study, the Airyscan was the best at this time. For the image analysis, particularly the evaluation of the phenotype of RNAi (Fig. 3a) and quantification of highly dynamic fusion (Figs. 2a, 3e), binarization was inevitable. In spite of the errors included in the binarization process, we were able to obtain enough information to explain the dynamics of circumferential cable formation. As for the orientation and elongation of the actin nanoclusters (revised Fig. 1f, Supplementary Figs. 1 b, c) and anisotropic motion (Fig. 5a, d), the other image analysis methods that do not require binarization (*OrientationJ* and *PIV analysis*) were also applied and have shown consistent results (revised Figs. 1h, i, Figs. 5b, c, e, f, respectively). Further analysis with fast live imaging with even higher resolution would be an issue in the future study.

To avoid overstatement regarding the size of the nanoclusters, the data were moved from main Fig. 1 e, f to Supplementary Fig. 1b, c. Also, the significant digits of the cluster sizes were reduced from 3 to 2, as written below.

Line 75: “As the process is highly dynamic in deep tissue and the size of the nascent actin cables was close to the diffraction limit, we used Airyscan which enables super-resolution imaging based on deconvolution and the pixel reassignment principle.”

Line 82: “The mode value of the cluster size estimated by elliptic fitting was about 140 nm on the minor axis and 210 nm on the major axis.”

- The authors mentioned that “Nanocluster fusion in the circumferential direction was preferentially stabilized; thus, the actin nanoclusters served as building blocks for cable formation”. However, from the images, it’s unclear whether the fusion of nanoclusters is stabilizing. In some cases, it may be stabilizing. However, as the visualization is contradictory, thus it is difficult to make a statement. For example, In Fig. 2a, cables #3 & 4 of the 0-min series, the cables are breaking down to nanoclusters and then reappear once

again, which makes it highly unstable. Maybe the authors could tone down the statement or give some more proof of the stabilization of the fusion process.

Thank you for the suggestion. The fusion and fission of the nanoclusters occur frequently at 0-min series, as you mentioned. Owing to the numerous repetitions, the slightly biased stabilization becomes apparent and reaches the steady state pattern, the periodic cable pattern, while maintaining the dynamic changes. Thus, it is difficult to show the stabilization within the several time-lapse images in Fig. 2a. The fusion sites were marked with magenta to make it clear. We rephrased the statement with tone down as written below.

Line 129: “because of the numerous repetitions of fusion and fission, the slightly biased stabilization becomes apparent and reaches the steady state pattern, the periodic cable pattern, while maintaining the dynamic local shape changes (Fig. 2c).”

- The authors said “the simulation demonstrated that the elliptical actin nanoclusters react to friction anisotropy and elongate in the direction of higher friction, developing into regular cables.” Although simulations show the biased orientation of the nanoclusters in the red case (+anisotropy + motor), I don't see any structure close to Fig. 1d in simulation results (Fig. 4b). The authors should mention the limit of insights from their model.

Because of the complexity of the model including multiple parameters and the limitation of the computational environment, a simulation takes 2 weeks to get the result. During the course of this study, we searched for parameter sets that meet the definition of the actin cable formation (dominant direction ≥ 75 , coherency ≥ 0.3), and found a condition indicated in Fig 4b, 4-M. Based on this parameter set, all of the rest of the simulation was performed. Thus, we are afraid to say that renewing the parameter set for the stripe pattern will take too long time to finish the revision in time. Also, there is a feedback mechanism between actin cables and the extracellular matrix to stabilize the structure, which has been reported previously. In this study, because we focused on the emergent process of the periodic pattern, and to address the generality of our finding, we did not implement the factor. The limitations of the model were described in the manuscripts and the related sentence was modified as follows.

Line 214: “Consequently, the fused actin clusters aligned toward the higher friction direction, forming a similar pattern with the periodic actin cables (Fig. 4b,c, 4-M, Supplementary Table 5).”

Line 324: “The actin self-organization mechanism was investigated by using an agent-based

model. In the simulation, the periodic cable-like structures appeared under the existence of a sufficient number of motors in addition to the filaments, crosslinkers and friction anisotropy. The simulation result (Fig. 4b, 4-M) partially reproduced the clear periodic cable pattern seen in tracheal cells at early Stage 16 (Fig. 1d). To improve to model, we might need to implement other factors for instance a feedback mechanism between actin cables and extracellular matrix to stabilize the cable structure, which has been previously reported¹¹.”

- The architecture of F-actins and cross-linkers in each nanocluster is not clear at all based on experimental results. Nevertheless, the model used arbitrary assumptions and reached a conclusion “Further analyses revealed that the clusters were formed through the competition between the effective diffusion of the filaments and their effective attraction due to the crosslinkers connecting the filaments.” How do you know whether the nanoclusters formed in the model will share the same origin with those in experiments?

Our theoretical analysis is based on a coarse-grained model of actin filaments, myosin II mini-filaments, and crosslinkers. The same approach with a similar set of components has been used successfully to reproduce dynamic actin structures (F. Ziebert and I. S. Aranson, Phys. Rev. E 77, 011918, 2008, J. A. Åström, P. B. Sunil Kumar, I. Vattulainen, and M. Karttunen, Phys. Rev. E 77, 051913, 2008). In-depth discussion on this model is provided in the revised supplementary information of the theoretical model, and in our previous publication (Phys. Rev. Res. 4, 043071 (2022)).

In this study, we used the coarse-grained model to study the mechanism behind the self-organization of the actin structures and the transition from the clusters to the cables. In fact, our model reproduced the clusters, the cables, and the labyrinth pattern, which resemble those observed in the experiment, by changing the parameters that are directly or presumably related to those changed in the experiment. This suggests that, although the model is very simplified compared to reality, it still captures the basic self-organization mechanism behind the structure formation observed in the experiments. As reviewer #2 suggested, comparing the detailed architecture of F-actins and cross-linkers in each cluster would provide further evidence of the relevance of the model. We would like to reserve them for future study.

Line 195: “Further analyses revealed that the clusters **in the numerical simulation** were formed through the competition between the effective diffusion of the filaments and their

effective attraction due to the crosslinkers connecting the filaments (Supplementary Information, Supplementary Fig. 6c).”

Line 272: “The simulation experiments omitting one or more of the three factors phenocopied the experimental results *in vivo*. Firstly, the simulation with the low number of crosslinkers (Fig 4b, 4-L) exhibited an isotropic pattern (Supplementary Table 3), which was similar to the downregulation of the crosslinker, *zasp52*, that showed orientation defect (Fig. 3a). Although the nanocluster fusion is stabilized anisotropically in *zasp52* RNAi (Fig. 3e), the weaker crosslinking might cause too frequent turnover of actin filaments and thus fail to maintain the anisotropic higher-order cable structure. Secondly, the absence of motors resulted in isolated clusters in the model. As the number of motors increases, the clusters are more connected to each other (Supplementary Fig. 6d2), consistent with the experimental results using *myosin II^{DN}* (Fig. 3b,f). Thirdly, when the system is isotropic, the direction of fusion is random resulting in a labyrinth pattern, which was comparable with the DAAM RNAi phenotype (Fig. 3f). Previous studies reported that formin family proteins sense the membrane tension and tensile stress to actin filament that positively regulate the F-actin polymerization^{31–34}. In tracheal cells, DAAM plays a critical role in nanocluster anisotropic motion and fusion anisotropy, making this formin protein a prime candidate for anisotropic membrane tension sensing. Altogether, the consistency of the experimental and simulation results strongly suggests the validity of the molecular machinery proposed in this study.”

- In their model (Fig. 4b), the authors found that the crosslinking density significantly affects cluster formation, and including the motor in the network enhances fusion, making the cable-like structure. However, it's still unclear whether motor density has any effect on producing cable-like structures.

In the simulation, the cable-like structures appeared under the existence of a sufficient number of motors in addition to the crosslinkers and the friction anisotropy. In fact, in the absence of motors, we obtained the isolated clusters of filaments and crosslinkers even under the existence of friction anisotropy. The clusters are more connected to each other as the number of motors increases, as shown in Supplementary Fig. 6d2. In addition, when the system is isotropic without friction anisotropy, although the clusters are connected to each other by the motors, their direction is random resulting in a labyrinth pattern. This argument was included in the discussion.

Line 208: “Myosin did promote the fusion of clusters and as the number of motors increases, the clusters are more connected to each other (Fig. 4b, c, 3-M, 3-H, Supplementary Fig. 6d2, Supplementary Table 3).”

Line 279: “Secondly, the absence of motors resulted in isolated clusters in the model. As the number of motors increases, the clusters are more connected to each other (Supplementary Fig. 6d2), consistent with the experimental results using *myosin II^{DN}* (Fig. 3b,f).”

- The number of data points in Fig. 1 (j-k) is insufficient to observe the trend clearly. Increasing the number of data points would result in a smoother curve and improve the visualization.

Our apologies for the insufficient number of data points. We doubled the number of data points by setting the interval of live imaging to 10 min and renewed the results (revised Figs. 1h, i, Supplementary Fig. 1g). It improved the temporal resolution and revealed that orientation was increased within a narrower time window, 40 min, than what we have stated (i.e. 60 min) before the revision. Thank you for your concrete advice. We also tried further increase of data points with 5 min intervals, but it caused photobleaching at later time points that compromised the accurate image analysis. The interpretation of the new results is written below.

Line 102: “The orientation increased over 40 min with high variation (-40 min to 0 min in Fig. 1h; time 0 corresponds to the completion of circumferential alignment of the actin pattern, Method *Others*), whereas the connectivity stayed low by -10 min (Fig. 1i). When the connectivity increased at 0 min, the variation of orientation substantially decreased (Fig. 1h, i). This measurement was supported by the skeleton analysis, which revealed that elongation of the actin nanoclusters became apparent at -20 min (Supplementary Fig. 1g).”

- In Fig. 2b and 2c, there is a mismatch between the color code and the mentioned direction. The blue color (0 deg-30 deg) should represent longitudinal, but in 2c, it is shown as circumferential. The figures should be made consistent to avoid confusion.

Thank you for the suggestion. We found that it was truly confusing, so the labeling was changed to the range of the orientation of each group (color code) in Fig. 2b and Fig. 3e.

- The type of simulations used in this study is not (coarse-grained) molecular dynamics simulations. It is more like an agent-based model or a discrete model.

Following the Reviewer's comment, we changed "coarse-grained molecular dynamics model" to "agent-based model".

REVIEWERS' COMMENTS

Reviewer #1 (Remarks to the Author):

The authors have addressed my comments. They added new experimental data to strengthen their conclusions. I cannot judge changes that were made to the interpretation of the simulations, or the conclusions based thereon.

The point about the developmental stage of appearance of periodic actin cables remains confusing. In Fig. 1c (top row, middle image, and close-up below) I can see periodic actin cables in a specimen labeled as "stage 15", but the text states that circumferential alignment appears by stage 16. Please clarify.

Line 60: What is meant by "High spatiotemporal imaging"?

Line 284: "Previous studies reported that formin family proteins sense the membrane tension and tensile stress to actin filament that positively regulate the F-actin polymerization." The meaning of this sentence is not clear. Please rephrase.

Reviewer #2 (Remarks to the Author):

The authors addressed all of my comments well. I recommend publication in Nature Communications.

Response to reviewer comments

Reviewer #1 (Remarks to the Author):

The authors have addressed my comments. They added new experimental data to strengthen their conclusions. I cannot judge changes that were made to the interpretation of the simulations, or the conclusions based thereon.

The point about the developmental stage of appearance of periodic actin cables remains confusing. In Fig. 1c (top row, middle image, and close-up below) I can see periodic actin cables in a specimen labeled as "stage 15", but the text states that circumferential alignment appears by stage 16. Please clarify.

We apologize for the insufficient description of the actin pattern formation during stages 15 and 16. To make it clear, we added a phrase in the manuscript.

Line 84: "As the development proceeds in stage 15, the clusters started to exhibit circumferential elongation (Fig. 1c, right), then by early stage 16, the circumferential orientation was completed with a statistically significant increase of the aspect ratio (Fig. 1d-f, Supplementary Fig. 1b, c, Supplementary Table 1)."

Line 60: What is meant by "High spatiotemporal imaging"?

We appreciate reviewer 1 for allowing us to make the expression more precise for the important point. We rephrased the sentence as written below.

Line 59: "The live imaging with high spatiotemporal resolution"

Line 284: "Previous studies reported that formin family proteins sense the membrane tension and tensile stress to actin filament that positively regulate the F-actin polymerization." The meaning of this sentence is not clear. Please rephrase.

To explain the previous studies clearly, we rephrased and added the sentences as below. Thanks to the suggestion, we think the discussion became more concrete.

Line 285: "Previous study reported that deformation of the cell membrane induces a rapid increase in cytoplasmic G-actin resulting in the positive regulation of F-actin polymerization by formin mDia1³¹. Also, pulling force applied to actin filament causes the increase of the directional polymerization of actin filament³²⁻³⁴. These mechanosensitive functions of formin

proteins in the cell cortex make DAAM a prime candidate for anisotropic membrane tension response in tracheal cells.”

Reviewer #2 (Remarks to the Author):

The authors addressed all of my comments well. I recommend publication in Nature Communications.